# Human mobility in a Bronze Age Vatya 'urnfield' and the life history of a high-status woman

Claudio Cavazzuti [1,2]*, Tamás Hajdu[3,4], Federico Lugli[5,6], Alessandra Sperduti [7,8], Magdolna Vicze[9], Aniko Horváth[10], István Major [10], Mihály Molnár[10], László Palcsu[10], Viktória Kiss[11]

1 Dipartimento di Storia Culture Civiltà, University of Bologna, Bologna, Italia, 2 Archaeology Department, Durham University, Durham, United Kingdom, 3 Department of Biological Anthropology, Eötvös Loránd University, Budapest, Hungary, 4 Department of Anthropology, Hungarian Natural History Museum, Budapest, Hungary, 5 Department of Cultural Heritage, University of Bologna, Ravenna, Italy, 6 Department of Chemical and Geological Sciences, University of Modena and Reggio Emilia, Modena, Italy, 7 Museo delle Civiltà, Sezione di Bioarcheologia, Rome, Italy, 8 University of Napoli "L'Orientale", Naples, Italy, 9 Hungarian National Museum, Budapest, Hungary, 10 ICER Centre, Institute for Nuclear Research, Debrecen, Hungary, 11 Institute of Archaeology, Research Centre for the Humanities, Hungarian Academy of Sciences Centre of Excellence, Budapest, Hungary

* claudio.cavazzuti3@unibo.it

**Data Availability Statement:** All relevant data are within the paper and its S1 File, S1–S5 Figs files.

**Funding:** This paper was supported by the Guest Researcher Fellowship granted by the Hungarian Academy of Sciences; by the Momentum Mobility

## Abstract

In this study, we present osteological and strontium isotope data of 29 individuals (26 cremations and 3 inhumations) from Szigetszentmiklós-Ürgehegy, one of the largest Middle Bronze Age cemeteries in Hungary. The site is located in the northern part of the Csepel Island (a few kilometres south of Budapest) and was in use between c. 2150 and 1500 BC, a period that saw the rise, the apogee, and, ultimately, the collapse of the Vatya culture in the plains of Central Hungary. The main aim of our study was to identify variation in mobility patterns among individuals of different sex/age/social status and among individuals treated with different burial rites using strontium isotope analysis. Changes in funerary rituals in Hungary have traditionally been associated with the crises of the tell cultures and the introgression of newcomers from the area of the Tumulus Culture in Central Europe around 1500 BC. Our results show only slight discrepancies between inhumations and cremations, as well as differences between adult males and females. The case of the richly furnished grave n. 241 is of particular interest. The urn contains the cremated bones of an adult woman and two 7 to 8-month-old foetuses, as well as remarkably prestigious goods. Using $^{87}Sr/^{86}Sr$ analysis of different dental and skeletal remains, which form in different life stages, we were able to reconstruct the potential movements of this high-status woman over almost her entire lifetime, from birth to her final days. Our study confirms the informative potential of strontium isotopes analyses performed on different cremated tissues. From a more general, historical perspective, our results reinforce the idea that exogamic practices were common in Bronze Age Central Europe and that kinship ties among high-rank individuals were probably functional in establishing or strengthening interconnections, alliances, and economic partnerships.

research project hosted by the Institute of Archaeology, Research Centre for the Humanities, Hungarian Academy of Sciences Centre of Excellence (Principal Investigator: Viktória Kiss) and by the grant from Hungarian Research, Development and Innovation Office, project number: FK128013 (Principal Investigator: Hajdu Tamás). The 14C measurements, conducted by the Atomki Laboratory, Debrecen were supported by the European Union and the State of Hungary, co-financed by the European Regional Development Fund in the project of GINOP-2.3.2-15-2016-00009 'ICER'.

**Competing interests:** The authors have declared that no competing interests exist.

## Introduction

The custom of cremation is a characteristic of the Bronze and Iron Ages in Europe. There are earlier examples, which appear sporadically and unsystematically across the continent [1–14], but in the second millennium BC there is a wide diffusion of large urnfields, often including hundreds—or even thousands—of burials [15–22]. In the central regions of the Danubian-Carpathian basin, following earlier Copper Age 'experiments', urn cremation becomes more formalized during the Hungarian Early and Middle Bronze Age (c. 2200-1500/1450 BC), largely coinciding with the historical development of the Vatya culture [23–26] (Fig 1).

Due to the fragmentary nature, and the consequently minor informative potential, of cremated remains, Vatya cemeteries have traditionally been investigated from a purely archaeological perspective (artefact typology, chronology, articulation of grave goods) [20, 26], with less attention paid to the human remains, the analysis of which is nonetheless essential for reconstructing funerary behaviours, demography, and social organisation [27]. As a result, while osteology, isotope analyses, and aDNA are dramatically expanding our view of the prehistoric populations that used inhumation, these aspects remain largely unclear for

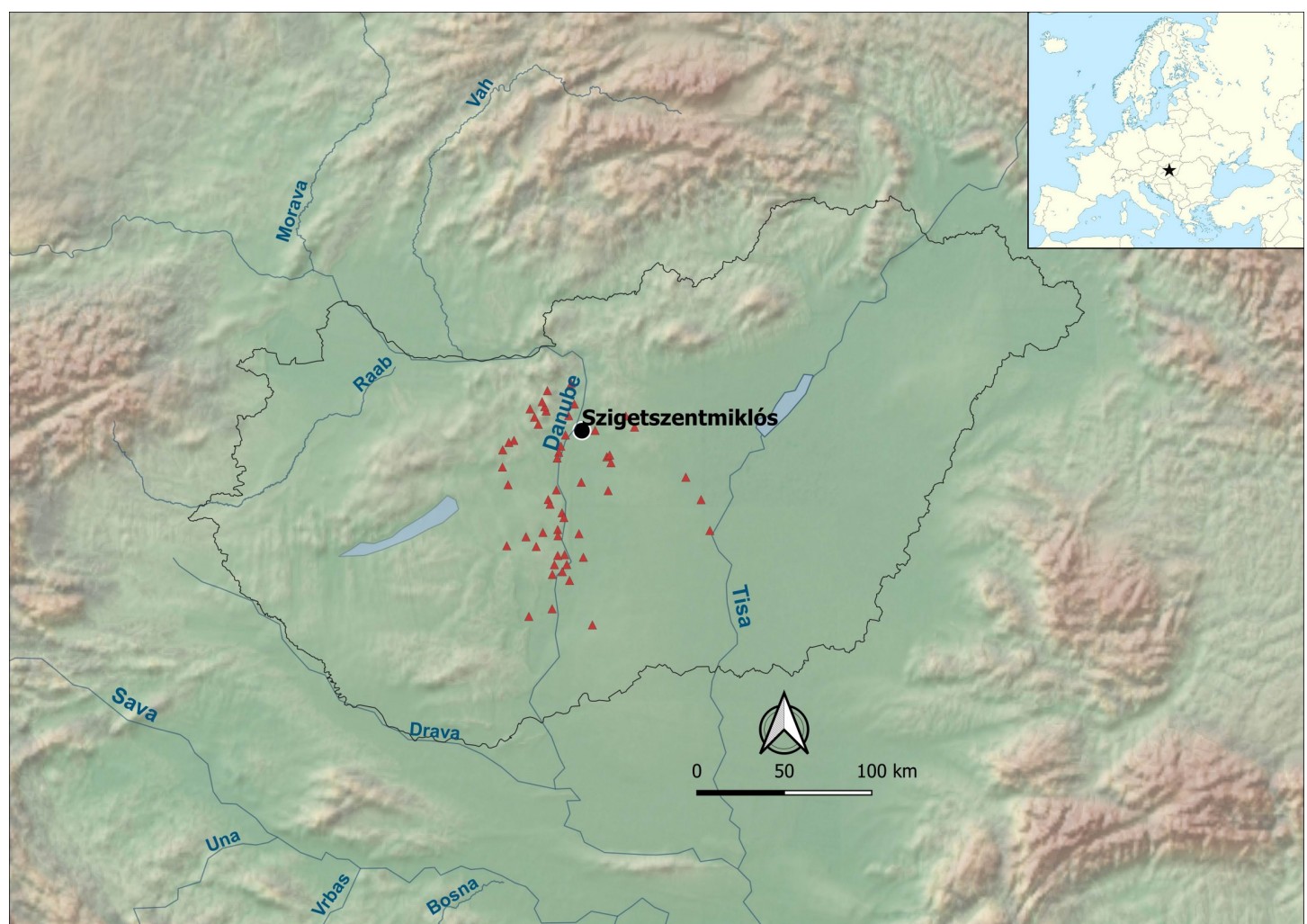

**Fig 1. Major Middle Bronze Age Vatya settlement sites along the Danube, in Central Hungary.** The map is constructed using "Natural Earth. Free vector and raster map data @ naturalearthdata.com" available at https://www.naturalearthdata.com/downloads/10m-raster-data/ and the location of the sites is taken from [51].

communities where cremation dominated, both in Hungary, and in other parts of Europe. Only very recently has bioarchaeological research undertaken the important challenge of filling this gap [e.g. 28–36]. Relevant to the present study, several new methodologies have been developed, which increase the accuracy of sex estimations [37–41], extend strontium isotope analyses to cremated materials, and verify the reliability of $^{87}Sr/^{86}Sr$ data on the petrous portion of the temporal bone [42–49]. In our study, we apply these new osteological methods and sampling strategies for the analysis of strontium isotopes to a sample of 29 burials from Szigetszentmiklós-Ürgehegy, one of the most important and best-preserved cemeteries of the Late Nagyrév and Vatya culture. Located a few kilometres south of Budapest, in the northern part of the Csepel Island, the cemetery includes over 500 urn cremations and eight inhumations [50].

Our main goals are *a)* contributing to the understanding of cremation practices of the Early and Middle Bronze Age in Hungary; *b)* exploring mobility patterns among inhumations and cremations, as well among individuals of both sexes and different age class, through strontium isotope analysis; *c)* applying a 'biographic' approach, namely targeting different human cremated tissues for strontium isotope analysis, to one particularly relevant case, that of burial n. 241, which contains multiple individuals and a variety of prestige grave goods. These latter have also been analysed from the point of view of geographical and contextual distribution, in order to integrate osteological and isotopic data in the broader archaeological framework.

## The Bronze Age 'urnfield' at Szigetszentmiklós-Ürgehegy

The cemetery at Szigetszentmiklós-Ürgehegy (c. 2150–1500 BC) represents one of the largest Middle Bronze Age urn cemeteries in Central Hungary. The site was uncovered during a rescue excavation preceding the construction of a major supermarket; an excavation led by M. Vicze found 525 burials within a 0.44-hectare area (Fig 2). The cemetery was situated on a slightly elevated sandy bank 500 metres from the Danube. Clusters of graves were identified, similarly to other major Vatya cemeteries [20]. The arrangement of the burials within these clusters was either linear or oval.

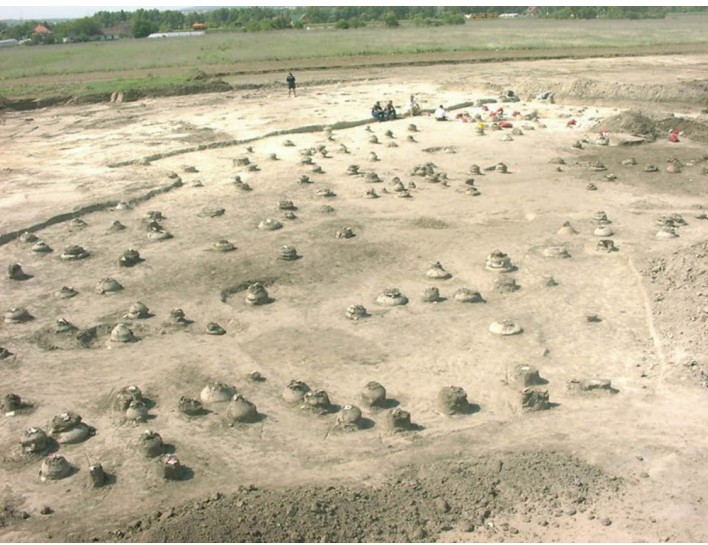 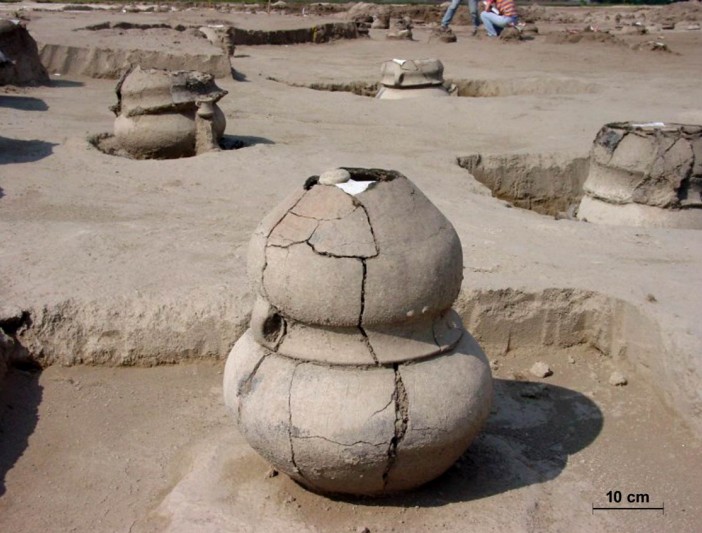

**Fig 2. Szigetszentmiklós-Ürgehegy 'urnfield' during excavation and a typical Vatya burial.**

A well-known but rarely documented custom of stone packing of graves and an occasional cluster of pebbles could be observed [20, 26, 52–54]. The petrographic analysis of the stones showed that, in some instances, they were brought to the site across the Danube, from as far as the northern Buda mountains, the pebbles are from the gravel of the river Danube [55].

The cemetery started to be used during the Early Bronze Age Late Nagyrév period (2150-2000/1900 BC) and continued to be active through the entire lifetime of the Vatya cultural unit, until the end of the Middle Bronze Age (1500/1450 BC). For the present study, the burials were divided into three major phases: early (1), middle (2), and late (3), where 'early' included graves from the Early Bronze Age and the Vatya I period (2150–1900 BC); 'middle' encompassed the Vatya II and III periods (1900–1700 BC); and the 'late' phase covered the entire (both early and late) so-called "Koszider" period of the culture (1700-1500/1450 BC) [20, 56]. The fundamental Vatya burial custom–integrating some Nagyrév elements [20, 56, 57]–consists of a cinerary urn with a small cup placed inside or beside it, and one or two bowls that cover or seal the mouth of the urn (Fig 2). Typically, the large vessel or urn was used to encompass the ashes, together with smaller ceramic vessels and other grave goods like bone and/or bronze ornaments. The urn was buried up to its neckline, covered with the bowl or bowls, and finally packed with stones, which probably served as grave markers. It is interesting to note that the Szigetszentmiklós-Ürgehegy cemetery has a higher number of "richly" furnished burials on average than other Vatya cemeteries known to date. The proportion of graves encompassing bronze and other status related grave goods usually varies between 5–10% in cemeteries where the entire chronological timespan of the culture is represented [20, 56]. However, our preliminary study of the Szigetszentmiklós material indicates that, in this specific case, the percentage could be as high as 25–30%. Significantly enough the cemetery at Szigetszentmiklós-Felsőtag, some 5 kms to the East on the other side of the Csepel island, also contains an unusually high number of burials with prestige goods. For example, there are two burials with neck-rings and an additional 21 with some extra adornments from a cemetery that has little more than 100 burials [20, 52, 56]. Another similarity between the two cemeteries is the occurrence of some specific decorations on vessels (textile-like impressions and handles with 'moustache' decoration) that are more frequent on the eastern side of the Danube, mainly in the territory of the Duna-Tisza Interfluve. Further, both cemeteries are located similarly: Ürgehegy 500 m, Felsőtag 800 m from the shore of the Danube on the almost exact opposite sides of the island. These shared traits, apart from implying a close relationship between the two communities, suggest the possibility that they could have been situated along an East-West route or line of network and/or a communication. Incidentally, the present-day highway (M0) skirting Budapest runs very close to and parallel with the natural line connecting the two sites. Traditionally, a change within the funerary custom of the local people, i.e. the introduction of inhumations [26], has been associated with the introgression of newcomers from the area of the Tumulus Culture in Central Europe around 1500 BC. It has been shown, however, that occasional inhumations were constantly present from the beginning of the Vatya culture [20]. However, most of the inhumations in Vatya cemeteries are without grave goods, hindering the identification of their date and possible origin. At Szigetszentmiklós-Ürgehegy there were eight inhumations, five of which had no grave goods.

## Materials and sampling strategy

We collected a total of 41 samples from different human tissues included in 29 burials (3 inhumations and 26 urn cremations; burial n. 241 includes three individuals, two of them analysed), as well as from 3 bone pins/needles from burials n. 215 and 241a (Table 1).

**Table 1. Number of analysed specimens for each type of sample.**

| | | |
|---|---|---|
| **Human** | Tooth enamel (from inhumations and subadult cremations) | 6 |
| | Tooth dentine (from inhumations) | 3 |
| | Cremated petrous bone | 24 |
| | Cremated femur cortical (Burial n. 241a) | 1 |
| | Cremated tooth dentine (Burial n. 241a) | 3 |
| **Artefact** | Bone pins/needles (cremated animal bone) | 3 |

Concerning inhumations, only three individuals had teeth sufficiently preserved to sample enamel and dentine for strontium isotope analysis. We sampled both dental tissues from first or second molars (M1, M2), the crowns of which develop at the age of 0–3 years and 3–8 years, respectively [58, 59]. It is well known that while enamel appears, in most cases, to be a reliable reservoir of biogenic strontium, dentine is affected by diagenesis and contamination from the burial environment [60]. In fact, the original strontium isotope composition of dentine may vary post mortem up to 100% during the biochemical exchange with the soil. Therefore, although we cannot measure the exact proportion of variation, dentine's values may be useful in a comparison to available strontium baselines for the area of the site.

Regarding cremations, the sampling strategy was carried out according to a set of criteria to ensure future analyses on the same contralateral anatomical element, and taking into account different archaeological and biological variables, namely sex and age-at-death of the individuals, burial rite (inhumation or cremation), quality/quantity of grave goods, and chronological phase of the burials. The petrous portion begins to form in utero at approximately 16–18 gestational weeks, is fully ossified by the time of birth, and does not undergo any subsequent remodelling after the age of two years [61–63].

Given that the petrous portion mostly forms before the end of weaning, its strontium isotope ratio should reflect the origin of the food consumed by the woman who breastfed the infant [62]. It seems scarcely probable that, in most cases, the infant's mother/wet nurse did not change over the duration of breastfeeding, did not undertake long journeys, and did not consume exotic foods, which are not documented from the Bronze Age archaeological record in the Central part of the Carpathian Basin [64].

The petrous portion has been proven suitable for strontium isotope analysis of cremated individuals, as the calcination of the bone increases its resistance to chemical diagenetic alteration, while not significantly changing its strontium isotope composition [42–44]. However, to re-test the reliability and consistency of the *pars petrosa* results, we contextually sampled and analysed the surviving first permanent molar (M1) crown of one subadult (burials n. 121); as both M1 enamel and bony otic capsule form at approximately the same age, their isotopic composition should not differ significantly. Two tests for verifying the reliability and consistency of the *pars petrosa* results, were performed in one former contribution [46] and gave positive results, thus confirming the possibility of estimating the individual's provenance also among cremations.

Except for burial n. 241, which included the cremated remains of an adult female (n. 241a) and of two foetuses of 28–32 gestational weeks (nn. 241b, 241c), all the other burials seemed to contain the remains of a single individual, since no exceeding bone was found in the assemblage. Moreover, the total weight of bone assemblages was always lower than the expectations for a complete cremation of a single individual reported in literature [65–70].

Burial n. 241 is of a particular relevance, as the urn also contains a golden hair-ring (*Noppenring*), a bronze neck-ring with flat-hammered ends (*Ösenring*, or *Ösenhalsring*), and two

ornamental bone pins/needles (*Knochennadeln*), which undoubtedly mark the individual's high-status.

The '*life history*' of n. 241 assumes even more importance when re-incorporated into the wider '*prosopography*' of Szigetszentmiklós-Ürgehegy individuals which, in turn, can be integrated into the broader, panoramic analysis of the mobility of people in the European Bronze Age [e.g. 45, 46, 71–74].

For this exceptional case, we applied a 'biographic' approach, collecting different dental and skeletal samples forming at different stages of the woman's life (Table 2. 1–5), in order to reconstruct her movements from childhood to the *pre mortem* period: the petrous bone, the dentine from M1, M2, and M3, and a cortical portion of the femur. There are three types of dentine: primary, secondary, and tertiary [75]. Tertiary dentine (also "reactionary" or "reparative dentine") forms primarily as a response to injury, such as caries and attrition [76] and may not be present, especially in the teeth of younger individuals. By contrast, primary dentine is formed during tooth development and remains unchanged throughout life. Once the roots are fully formed, secondary dentine is progressively laid down on the walls and roof of the pulp chamber, gradually reducing its size [77]. The process starts at early stages (with differences between teeth) and is regular during the entire life course of an individual, even though it may be partially influenced by dental caries or other physical/chemical damage to the tooth. Thus, secondary dentine can preserve discrete or cumulative signals (also the chemical ones), of events and long-term processes, respectively. The latter occurs over a period spanning from root completion to tooth shedding or to the death of the individual, i.e. as long as the tooth remains vital [78–80]. Therefore, the roots, for their mechanism of addictive deposition of primary and secondary dentine, are a key potential source of information for the present analysis. Moreover, a recent study by Taylor et al. has also demonstrated that cremated dentine samples are not prone to contamination by strontium from the burial environment, and may provide accurate strontium isotope results that can be used for the analysis of provenance [47].

The adult femur combines the isotopic signal of the formation phase and an average of the last years of life. In fact, once growth is complete, the bone is continuously remodelled through a cycle of resorption and apposition. The rate of bone remodelling is estimated to be from 3 to 8% per year, and this rate decreases with advanced age, with differences between skeletal parts [81–83]. We also sampled a few milligrams of bone from the *pars petrosa* of one of the two foetuses (burial n. 241b), which were forming in utero in the last days of the woman's life

**Table 2. Samples taken from burial 241 for $^{87}$Sr/$^{86}$Sr analysis and period of formation of the dental/skeletal element.**

| Sample | Period of formation | Remodelling *in vita* |
|---|---|---|
| 1) Woman's (241a) petrous bone | in utero | Negligible |
| 2) Woman's (241a) M1 root | ~3–9 years old | Continuing deposition of secondary dentine throughout life after the eruption of the tooth |
| 3) Woman's (241a) M2 root | ~8–13 years old | Continuing deposition of secondary dentine throughout life after the eruption of the tooth |
| 4) Woman's (241a) M3 root | ~15–20 years old | Continuing deposition of secondary dentine throughout life after the eruption of the tooth |
| 5) Woman's (241a) femur cortical | ~last 10–20 years of life | Yes |
| 6) Child's (241b) petrous bone | ~last days/weeks of the woman's life | No |
| 7) Two pins/needles (made of animal bone) | - | - |

(Table 2. 6). Additionally, we collected two samples from the two pins/needles (made of animal bone), which adorned the funeral dress of the woman on the pyre. As the bone pins/needles are calcined, we can exclude a significant impact of the diagenesis on the chemical composition of the artefacts.

Sex and age class of the individuals were determined using published morphological and metrical variables [36, 37, 40, 84–91]. Considering the objective difficulties in assigning sex to cremated individuals, we integrated the observations of dimorphic anatomical traits with osteometric parameters developed for Italian Bronze Age and Iron Age cremated series [37], which substantially increases the accuracy of sex estimations in the case of proto-historic populations.

The chronological phase of each burial was identified through relative chronology based on artefact typology of the grave goods such as urns, bowls, additional vessels and bronze artefacts of the Szigetszentmiklós-Ürgehegy cemetery, which have been broadly discussed in the wider context of Vatya culture absolute and relative chronology [20]. Seven of the 29 burials belong to "phase 1", displaying the distinctive characteristics of the Late Nagyrév–Early Vatya transitional traits. These are typically the finger-impressed bands on the urns (nn. 208, 243, 459, 466); short-curved neck (n. 459); Nagyrév incised geometric lines (n. 241); Nagyrév small vessel with Vatya urn (n. 242); period specific bronze ornaments like neck-ring, panpipe-shaped, small rectangular or trapezoid sheet ornaments (nn. 241, 279), the gold hair-ring of n. 241 also belongs to this timeframe. Parallels and examples for these attributes can be seen in all early Vatya cemeteries like Szigetszentmiklós-Felsőtag, Kulcs, Biatorbágy, and the relevant phase of Dunaújváros-Dunadűlő with further references [20, 26, 52, 92]. Ten burials are representing the so-called Vatya II and Vatya III features that have been labelled "phase 2". The most common elements of this phase are: narrow long neck (n. 121); or funnel like neck from globular to biconical body usually with one long handle on the neck of the urn (nn. 189, 212, 215, 427, 440); the decrease and eventually the almost complete lack of decoration on the vessels is a typical aspect of this period. Bronze finds like the burnt dagger (n. 189) or the poppy-headed pins (nn. 215, 480) and various (bone and faiance) small beads (nn. 215, 350) are widely in use during this period. Similar vessel types and bronze object come from other Vatya sites like Dunaújváros, Budatétény etc. [20, 26, 93]. "Phase 3" comprises nine burials that could be identified as Early and Late Koszider period [53] primarily based on the ceramic styles and decorations. The flaring rims and specific boss decorations both on the bowls and urns (nn. 11, 317, 433, 449) are the most general indicators of the changing times that mark the advent of the Koszider period. The conical shaped bowls with inverted rims that are ornamented with larger horizontal knobs is an invention of this period (nn. 117, 317, 515). Next to the cone-headed pin from grave n. 224 that can be dated to this phase, there are other non-period specific bronze objects like buttons, spiral tubes, spectacle pendants that came to light from graves of all periods.

In absolute chronology, based on AMS dating of collagen samples from tell settlements, "phase 1" represents the period of Late Nagyrév and Early Vatya traditionally dated between 2150 BC and 1900 BC, "phase 2" ranges between 1900 BC and 1700 BC, and "phase 3" between 1700 BC and 1500/1450 BC [94–96]. Some burials, due to the absence of grave goods or non-datable ceramic types, and also lacking radiocarbon dating, were left chronologically undetermined ("N.A."). The "phase 3" in Szigetszentmiklós-Ürgehegy cemetery was dated between 1750 and 1430 BC, correlating with other absolute dates of the Koszider period.

Radiocarbon dates have been produced for 11 burials (nine on the bone apatite of cremations and two on bone collagen of inhumations). Since dates on cremated bones are evidently incompatible with the relative chronology given by grave goods analysis, we reported and discussed the data in the Supplementary materials (S1 File). The two inhumations (nn. 190 and

489), however, whose dates have been obtained by the analysis of bone collagen, yielded 1550–1290 cal. BC (2σ) and 1690–1500 cal. BC (2σ), respectively. These dates can be ascribed to the "phase 3", of Vatya culture.

The three grave good classes indicate the relative "richness" of the burials and were determined as follows:

- Grave good class 1: individuals with prestige goods (gold hair-ring, bronze neck-ring, bronze dagger and/or large number, i.e. >30 pieces of bronze plate ornaments and/or pendants)

- Grave good class 2: individuals with few common bronze ornaments (<30 pieces of bronze plate ornaments and/or bronze sheet pendants, buttons, pins, imported or other special ceramic vessel)

- Grave good class 3: individuals with ceramic grave goods only.

## Methods: Strontium isotope analysis

Strontium isotope ratios in odontoskeletal remains are commonly employed to track individuals' mobility in different stages of their lives [97]. Radiogenic strontium-87 ($^{87}Sr$) originates over time from the radioactive decay of rubidium-87 ($^{87}Rb$; half-life of 48.8 Ma), while strontium-86 ($^{86}Sr$) is stable. Their ratio ($^{87}Sr/^{86}Sr$) is therefore dependent on the age of a given bedrock, but also on its geochemical nature (e.g. the initial Rb content). Older/crustal geological units generally display more radiogenic $^{87}Sr/^{86}Sr$ values, while younger/mantle materials show lower ratios. The sediments that form alluvial plains reflect the ratio of their parent material, or an admixture of the ratios that characterise the different geological units affected by the erosive activity of rivers in the uplands. Soluble strontium then enters the food chain, first absorbed by plants, and subsequently fixed in bone and tooth bioapatite of animals/humans by replacing calcium [97].

The possible provenance of individuals is estimated by comparing the ratio between strontium-87 and strontium-86 in bones/teeth with the local baseline values measured in faunal/vegetal samples, as well as soils and waters, from the burial site or its geologically coherent immediate hinterland. If environmental samples are not available, or contaminated by modern sources [e.g. 98], archaeological fauna and human bone/dentine can be used to support local baselines, although, especially the latter, is not an ideal option and must be considered with caution, as diagenesis may alter to variable degree original strontium signatures. When $^{87}Sr/^{86}Sr$ values measured on one individual are similar to the local baselines, the subject is plausibly indigenous; when they differ, the individual certainly originated in a different place [99]. This technique has been in use for several decades and is now a common tool in mobility studies [100].

For our analysis, we followed well-established standard procedures. Tooth enamel was mechanically abraded from the surface with a dental burr and the removed material discarded. Any adhering dentine was then removed and a resulting clean core enamel of approximately 20 mg isolated for strontium isotope analysis. Concerning cremated individuals, petrous portions were sampled applying Jorkov et al.'s method [101]: the petrous bone was drilled at a 90˚ angle into the otic capsule (0.5–0.8 cm of depth), between the internal acoustic meatus and the subarcuate fossa with a low speed (2-mm diameter) drill, and a small fragment of bone was isolated. Calcinated bone and tooth specimens were cleaned following the protocol described by Snoeck et al. [43]. Bone and dentine from burial n. 241 were sampled through mechanical abrasion of the root surface and detaching approximately 20 mg fragments for isotopic analysis. In detail, each sample was treated with 1M acetic acid for 3 minutes in an ultrasonic bath,

followed by three washes with MilliQ. Enamel samples were rinsed with MilliQ in an ultrasonic bath.

Samples were digested in 6M $HNO_3$, dried down and re-digested in 3N $HNO_3$. 30µl columns filled with Eichrom Sr-spec resin were employed for the subsequent chromatographic separation of Sr. The whole procedure was conducted in the clean lab of the Department of Chemical and Geological Sciences (University of Modena and Reggio Emilia), with a Sr blank typically lower than 100 pg. In this specific case, Sr laboratory blank was below 60 pg. Sr isotope ratios were measured using a Neptune (ThermoFisher) multi-collector inductively-coupled-plasma mass spectrometer (MC–ICPMS) housed at the Centro Interdipartimentale Grandi Strumenti (University of Modena and Reggio Emilia). Seven Faraday detectors were used to collect signals of the following masses: $^{82}Kr$, $^{83}Kr$, $^{84}Sr$, $^{85}Rb$, $^{86}Sr$, $^{87}Sr$, $^{88}Sr$. Sr solutions were diluted to ~30 ppb and introduced into the Neptune through an APEX desolvating system. Corrections for Kr and Rb interferences follow previous works [e.g. 102]. Mass bias corrections used an exponential law and an $^{88}Sr/^{86}Sr$ ratio of 8.375209. The Sr ratios of samples were reported to a NIST SRM 987 value of 0.710248 [103]. NIST SRM 987 yielded an average $^{87}Sr/^{86}Sr$ ratio of 0.710235 ± 0.000019 (2 S.D., n = 17).

To statistically detect possible non-local individuals, we employed two approaches: the median absolute deviation from the median (MAD) and the Tukey's fences method. Concerning the MAD, we calculated the 3MADnorm as reported by Lightfoot and O'Connell for oxygen isotope analysis [104] and Leys et al. [105], namely multiplying by 3 the MAD and scaling it to a b value. This latter is usually assumed as a constant (1.4826) for normal distributions, neglecting outlier-induced abnormalities [105]. Excluding outliers though a Grubb test (p < 0.05; 0.71209 is an outlier), our data appears as normally distributed (Shapiro-Wilk test; W = 0.954; p = 0.105), hence a b value = 1.4826 has been used as scaling factor for the MAD. Tukey's interval has been calculated as Q1-1.5(IQR) and Q3-1.5(IQR), where IQR is the interquartile range [106]. A two-tailed T-test was also performed searching for group differences between cremated vs. inhumated individuals. We acknowledged that the small number of inhumated individuals (n = 3) may impact the prediction power of the test. All the tests have been performed manually using either Microsoft Excel or MATLAB.

All necessary permits were obtained from the Ferenczy Museum Center, Szentendre, for the described study, which complied with all relevant regulations.

## Methods: Geology of Central Hungary area and biologically available strontium 'isoscape'

Most parts of the Great Hungarian Plain are covered with Quaternary, loose, clastic sediments deposited by alluvial events from the Danube and Tisza rivers and their tributaries [107, 108]. The surface units are predominantly made up of fluvial sediments, which are composed of cyclically alternating gravel, sand, silt and clay. The Upper Pleistocene loess prevails on the mountain margins of the basins of the Great Hungarian Plain and on the Mezőföld, while the Upper Pleistocene, Holocene wind-blown sand mostly occurs in the Danube–Tisza Interfluve. The characteristic formations of the Transdanubian Hills are Late Miocene sediments, loess, and fluvioeolian sand.

The western bank of the Danube in the Buda area is part of the SW–NE formations of the Transdanubian Range that stretches from the Keszthely Mountains to the Pilis–Visegrád Mountains and is mostly made up of Triassic sedimentary formations (marine dolomite and limestone). Older, Palaeozoic formations outcrop from below the Triassic successions around the Balaton Highlands, and the Balatonfő and Velence Mountains, while younger, Mesozoic formations cover larger areas in the upland regions of the Bakony and the Gerecse. From the

Visegrád to the Tokaj Mountains, the North Hungarian Range is mostly formed by Miocene volcanic rocks, with the significant exception of the Bükk and Uppony Mountains, which show much more complex and older geology, with Palaeozoic, Triassic and Jurassic marine sediments.

In a recent publication, Giblin and collaborators [109] considered the overall geology of the Danubian-Carpathian basin, presenting a first 'isoscape' based on a limited, but significant set of data. Since then, other research projects have enriched the biologically available strontium baselines [110–117]. We collated all the published isotopic signatures and generated a new georeferenced 'isoscape', which differentiates the various baseline sources, namely modern plants, archaeological bone/dentine, archaeological fauna, river/mineral waters (Fig 3). As shown on the map, most of the isotopic data come from Transdanubia, Balaton lake area, the Körös region and the south of Hungary. By contrast, the Budapest hinterland has been less intensively investigated. In the $^{87}Sr/^{86}Sr$ analysis of Hungarian Bell Beaker burials, Price et al. reported bone values ranging from 0.7093 to 0.7097 at Budapest-Békásmegyer and 0.7094 to 0.7100 at Csepel Vízcsőárok and Szigetszentmiklós-Üdülősor [118]. However, as only one individual shows a value of 0.7100, a range of 0.7093–0.7097 seems more reasonable.

In another recent work, Giblin and collaborator analysed 17 archaeological samples of animal tooth enamel from Érd-Hosszúföldek Bronze Age site, which is located around 10 km west of Szigetszentmiklós on the opposite bank of the Danube. Strontium isotope values range from 0.7090 to 0.7097 [119].

Budapest and the Csepel Island have been the most heavily cultivated and now urbanised areas of Hungary. The few cultivated fields are very close to houses and factories and have been heavily manured in the recent past, as well as today. The area around the site is heavily affected by anthropogenic activity and, to elude the impact of modern contamination [e.g. 98, 120], we avoided sampling modern plants to build baselines for Szigetszentmiklós-Ürgehegy. Since no animal bones have been recorded from the cemetery so far, we integrated published baselines with human bone, dentine, and 0-5-year-old infant tooth enamel from Szigetszentmiklós-Ürgehegy and from the nearby cemetery of Százhalombatta-Belső-Újföldek as further support for local $^{87}Sr/^{86}Sr$ signatures. We concentrated on children's teeth because several larger, previously published series of Sr isotope data confirm significantly less variable $^{87}Sr/^{86}Sr$ ratios among the teeth of children than among those of adult individuals; as young children are unlikely to have migrated, their values usually reflect the local signatures [45, 46, 71, 72, 114, 117, 121].

## Results of the osteological analysis and focus on burial n. 241

The skeletal remains from the three inhumations were in a poor state of preservation. The few preserved teeth and long bone fragments could nonetheless be attributed to adult individuals of undeterminable sex.

The cremated bones of the 26 analysed burials showed a rather homogenous, calcined aspect, which indicated a relatively high temperature reached by the pyre (>800˚C) [28, 69, 122–125]. With the exception of n. 241, all the other 25 urns seemed to contain the remains of a single individual. In the overall sample, seven adults showed typical masculine morphologic and metric traits, 11 adults could be estimated as female, two individuals were of undetermined sex, two non-adults were 5–10 years of age and four were 2–5 years of age. The only individuals younger than 2 years were 241b and 241c, namely the foetuses associated with the adult female 241a.

After the osteological analyses, sex estimations were compared to indicators of gender, based on grave goods (if present). A complete correspondence between (osteologically

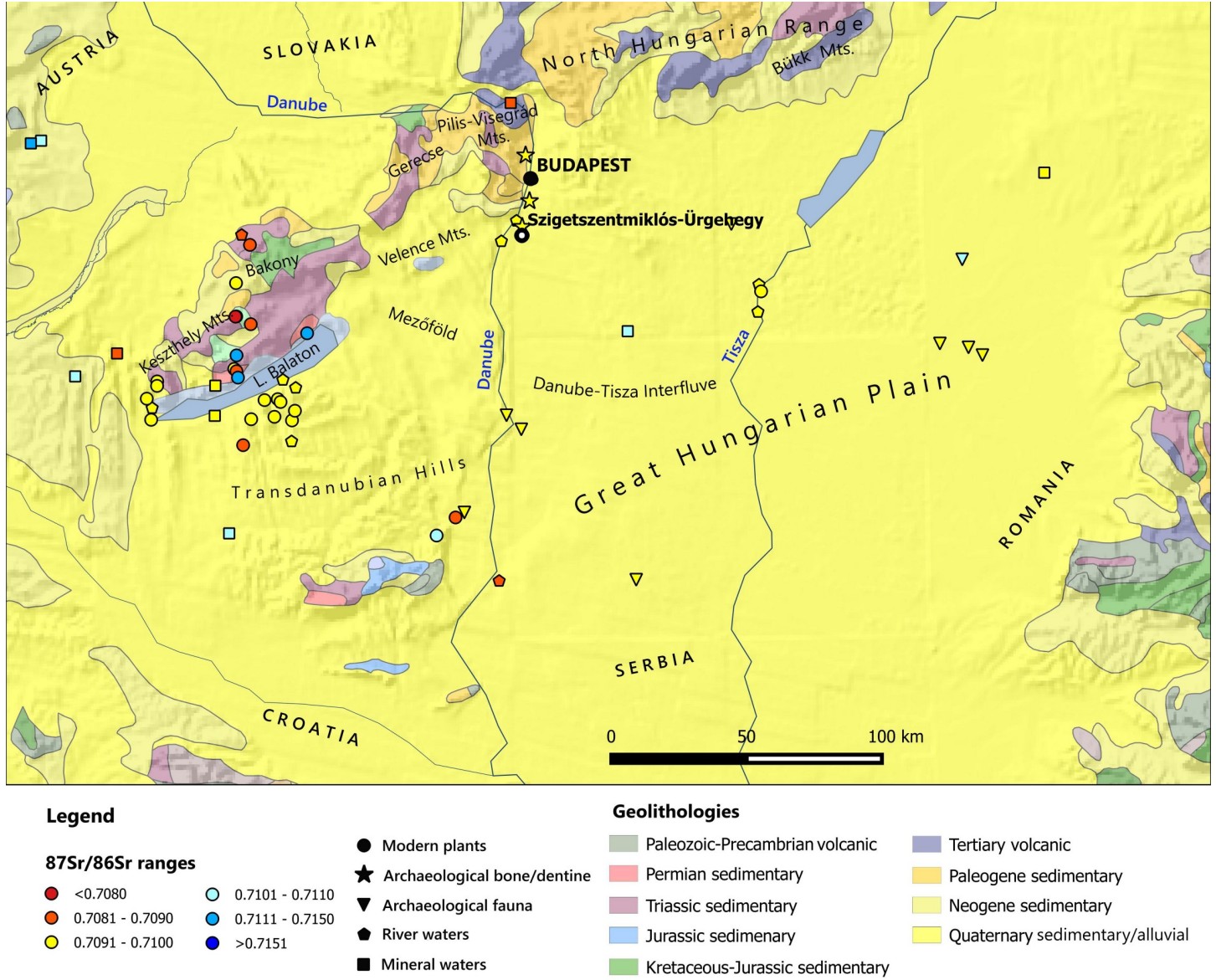

**Fig 3. Location of Szigetszentmiklós-Ürgehegy cemetery and of the biologically available strontium baselines reported in literature [109–118].** The geological map is constructed by using public domain wms data downloadable from https://certmapper.cr.usgs.gov/data/apps/world-maps/, which are of Public Domain. Credit: U.S. Geological Survey, Department of the Interior/USGS, U.S. Geological Survey.

estimated) females and typically feminine grave goods was found. As Vatya male burials are usually not equipped with weapons or typically masculine objects [20, 126], such a test could not be conducted on male individuals.

Given that the urns were not damaged by modern ploughing/earthworks, the weight of cremated bone recovered plausibly reflects the original amount placed in them before the final deposition in the pit. S1 Fig shows the distribution of cremated bone weights of adult males, females, and infants. The mean weight was 1584 g for adult males (st. dev. = 346 g), 1000 g for adult females (st. dev. = 531 g), 675 g for 5-10-year-old infants (st. dev. = 305 g) and 264 g for 2-5-year-old infants (st. dev. = 256 g).

**Table 3. Relative weight (in %) of the anatomical districts for each category of individuals.** Two adults of undetermined sex (n. 208, 433) are not included.

| | Number of samples | Descriptive statistics | Bone weight (g) | % cranium | % long bones | % thorax, pectoral girdle and axis | % pelvis | % hands/ feet | % NA |
|---|---|---|---|---|---|---|---|---|---|
| Adult males | 7 | mean | 1584 | 8.7 | 24.5 | 8.8 | 0.7 | 0.8 | 56.4 |
| | | std. dev. | 346 | 2.4 | 10.5 | 11.4 | 0.6 | 0.7 | 7.7 |
| Adult females | 11 | mean | 1000 | 11.2 | 27 | 5.3 | 0.5 | 0.7 | 55.3 |
| | | std. dev. | 531 | 7.4 | 8.2 | 4.1 | 0.7 | 0.9 | 17.2 |
| Infants | 6 | mean | 428 | 10.1 | 13.9 | 2.7 | 0.1 | 0.1 | 73.1 |
| | | std. dev. | 327 | 6.7 | 12.2 | 2.3 | 0.2 | 0.1 | 17.1 |

The bone assemblage normally included all anatomical areas, although cranium, long bones, and thorax/pectoral girdle/vertebral column showed higher representation when compared to pelvic and hand/foot bones. A significant quantity of small fragments could not be unambiguously assigned to a specific anatomical region. The relative weight of each anatomical area among adults of both sexes and subadults is shown in Table 3.

As stated above, burial n. 241 showed very special characteristics. Most bones (1516 g) could be attributed to an adult subject (241a; Fig 4-left), with an age-at-death of between 25 and 35 years. The epiphyses of the long bones and clavicles, as well as the iliac crest, were completely fused, the cranial sutures showed no trace of obliteration and no degenerative markers were visible on the vertebrae, or on other articular joints.

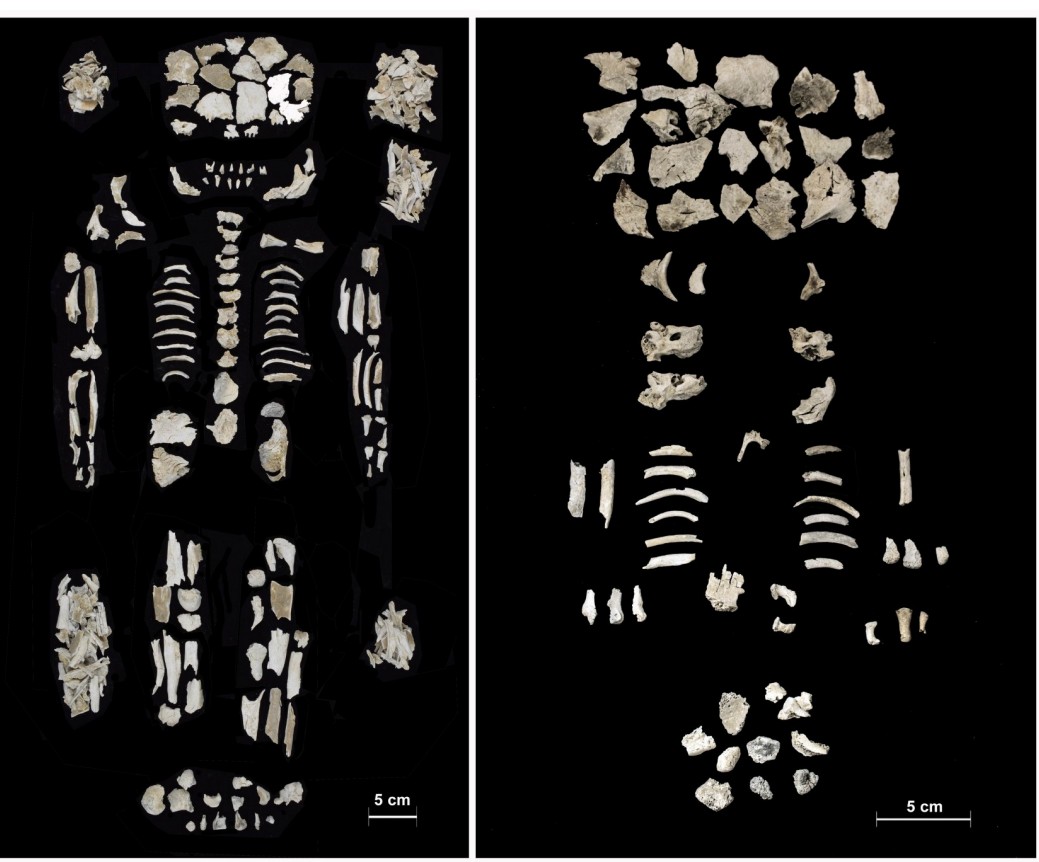

**Fig 4.** Left: Bone assemblage from burial n. 241a (adult female individual). Right: Bones attributable to both foetuses (n. 241b and 241c).

**Table 4. Comparison between the measurements taken on the sexual dimorphic anatomical traits of the burial n. 241a compared with the cut-off points between female and male metric distributions for each trait published by Cavazzuti et al. [24].** Only the medio-lateral diameter of the first metatarsal is more robust than the cut-off point. All the other traits are more compatible with metric distributions for females.

| Anatomical trait | Measurement on 241a (mm) | Cut-off point for sexual dimorphism (mm)–after Cavazzuti et al. [24] |
|---|---|---|
| Axis: dens anteroposterior diameter | 9.41 | 9.55 |
| Axis: dens transverse diameter | 8.08 | 9.10 |
| Humerus: trochlea minimum diameter | 10.84 | 13.28 |
| Radius: head maximum diameter | 17.24 | 18.32 |
| First metatarsal: dorso-plantar diameter | 15.07 | 16.17 |
| First metatarsal: medio-lateral diameter | 18.53 | 17.02 |

The female sex was revealed by the overall gracility of the skeleton, as well as by morphological and metric traits. These latter were evaluated using the discriminant cut-off points for sexual dimorphism developed in a study by Cavazzuti et al. [37]. Specifically, the mandibular ramus was visibly inclined and its posterior margin almost rectilinear; the mandibular condyle appeared very gracile; the occipital was very smooth with the protuberance barely marked; the greater sciatic notch appeared very wide with a deep and well-defined preauricular sulcus. Concerning metric evaluations, of the six anatomical traits which were enough preserved to be measured, five were particularly gracile and fell below the cut-off points determined by Cavazzuti et al. (Table 4). As these cut-off points represent the statistical limits that discriminate between female (more gracile) and male (more robust) metric distributions for each anatomical trait, our measurements support the estimation of the female sex for the individual 241a.

The two foetuses 241b and 241c were identified from the gracility and dimensions of a good number of fragments (44 g; Fig 4-right), which evidently differed from the majority bones attributable to the adult individual (1516 g) and from the presence of three exceeding pars petrosals (two right and one left, very similar in size). The best preserved of these three bones measured 21.1 mm in length. Given that a small part of the postero-inferior point was broken, we could estimate an original total length of around 23 mm, which would point to approximately 29–30 gestational weeks [85, 127, 128]. However, since the two foetuses were twins and therefore potentially characterised by a smaller body size *in utero*, and taking into account bone shrinkage due to the cremation rite, we may assume a higher age at death, approximately 32 gestational weeks (7–8 months of gestation). We had no evidence to indicate whether the childbirth actually occurred, or the foetuses died *in utero*, as consequence of the woman's death.

The urn also contained ornaments, which were part of the deceased's dress: a bronze neck-ring (*Ösenring*), a gold hair-ring (*Noppenring*) and two cremated and fragmented bone pins/needles (*Knochennadeln*), found commingled with the human remains (Fig 5). These latter were also found in grave n. 215 (adult female). All these objects are typical of the Early Bronze Age in Central Europe, in the vast region that comprises the Danube and the Carpathian basins. Their distributions will be discussed in the following paragraphs in relation to the strontium isotope data.

## Results of the $^{87}$Sr/$^{86}$Sr analysis

Our baseline data largely support the ranges published by Price et al. and Giblin's et al. and point to a site baseline of 0.7091–0.7095. Taking the two studies altogether, we may estimate Budapest's immediate hinterland signatures ranging from 0.7090 to 0.7097, with the exception of the Buda, Pilis-Visegrád and Gerecse Mountains, which would probably yield different values as a consequence of their older geology (Fig 3).

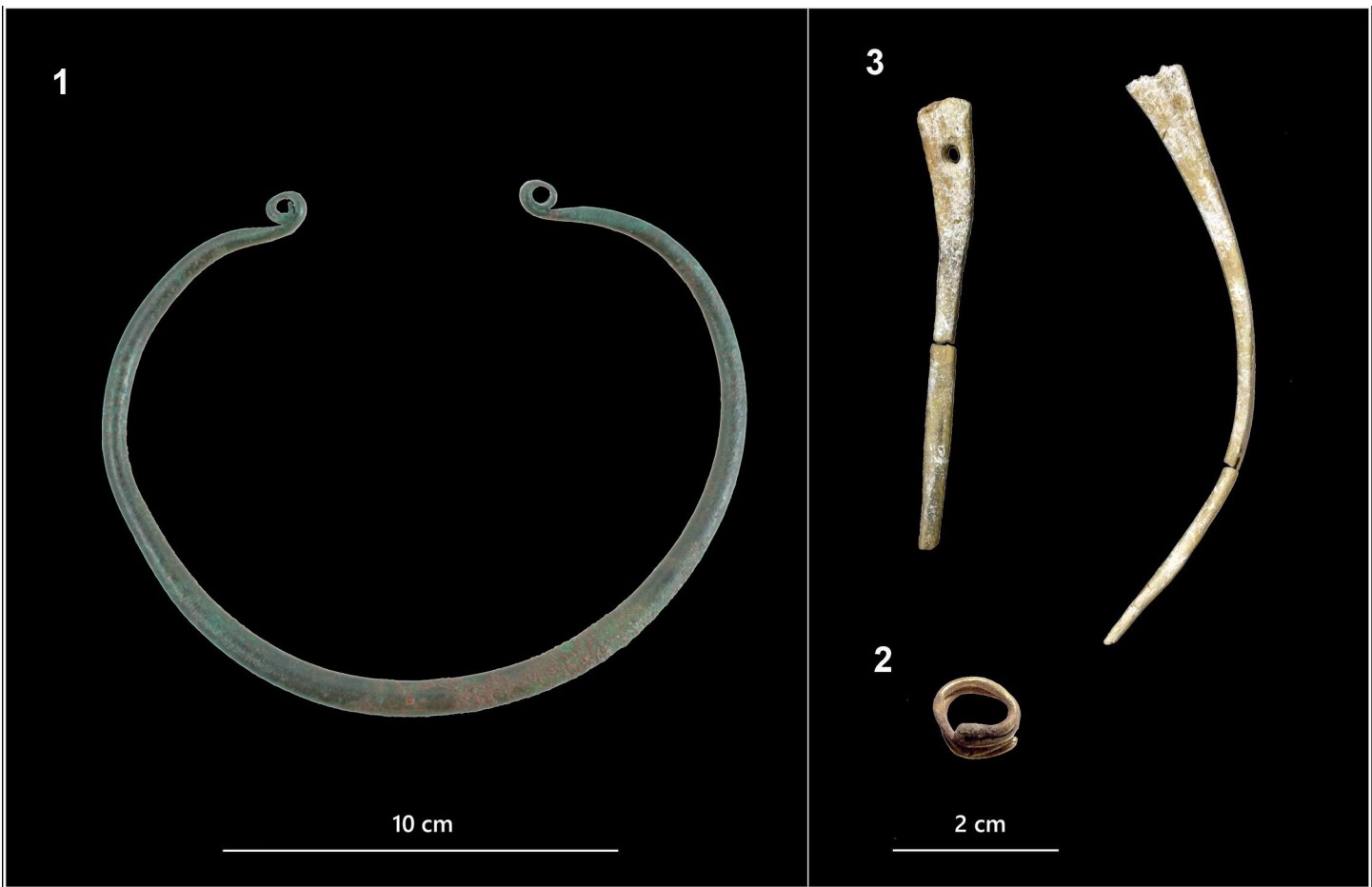

**Fig 5. Grave goods from burial n. 241.** 1. Bronze neck-ring (*Ösenring*); 2. Gold hair-ring (*Noppenring*); 3. Bone pins/needles (*Knochennadeln*).

Overall, strontium isotope compositions of humans varied from 0.70888 to 0.71209 (Table 5). Most of the individuals (N = 17; 68.6% of the sample) are concentrated in a narrow range between 0.70908 and 0.70918. Another cluster ranges from 0.70924 to 0.70956 (N = 9; 31% of the sample; Fig 6). Although these variations might appear small in absolute terms, relatively to the great geolithological uniformity of the area where the site is located, slight differences can be regarded as meaningful. Statistical results are summarized in Table 6. Both employing 3MADnorm or Tukey's IQR [104–106], only two individuals are detected as outliers, namely n. 215 ($^{87}$Sr/$^{86}$Sr = 0.70956) and n. 243, the most radiogenic of the dataset ($^{87}$Sr/$^{86}$Sr = 0.71209). N. 215, however, fall within the local strontium signatures (Szigetszentmiklós = 0.7091–0.7095; immediate hinterland = 0.7090–0.7097). Individual n. 241a (the lowest radiogenic; $^{87}$Sr/$^{86}$Sr = 0.70888) is outside the local range but lays at the lower edges (but still within) of the statistical intervals (3MADnorm or Tukey's IQR). Hypothetically n. 241a might be considered non-indigenous, but such interpretation needs further analysis, both from a biogeochemical and archaeological perspective. In fact, while n. 243 does not contain any grave goods at all, n. 241a is equipped with a set of prestigious items, which allows us add further elements to the interpretation. Moreover, we sampled other dental/skeletal tissues from n. 241, in order to analyse $^{87}$Sr/$^{86}$Sr variations during her various life stages.

Isotope results for the 26 cremations range from 0.70888 to 0.71209, while the analysis of tooth enamel from the three inhumations (nn. 190, 476, 489) yielded values between 0.70917

**Table 5. Results of the analyses.**

| Grave N. | Rite | Sex | Age | Phase | Grave good complexity | Type of sample | ⁸⁷Sr/⁸⁶Sr | 2se |
|---|---|---|---|---|---|---|---|---|
| 11 | C | F | 25–40 | 3 | 3 | petrous portion | 0.70915 | 0.00001 |
| 117 | C | M | 40+ | 3 | 3 | petrous portion | 0.70925 | 0.00001 |
| 121 | C | U | c. 2 | 2 | 2 | petrous portion | 0.70913 | 0.00001 |
| | | | | | | M1 enamel | 0.70924 | 0.00001 |
| 189 | C | M? | 20–40 | 2 | 1 | petrous portion | 0.70913 | 0.00001 |
| 190 | I | U | 30–40 | NA | 3 | M1 enamel | 0.70917 | 0.00001 |
| | | | | | | M1 dentine | 0,70907 | 0,00001 |
| 208 | C | U | 20–40 | 1 | 3 | petrous portion | 0,70909 | 0,00001 |
| 211 | C | M | 20–40 | 2 | 3 | petrous portion | 0,70907 | 0,00001 |
| 212 | C | F? | 25–40 | 2 | 3 | petrous portion | 0.70947 | 0.00001 |
| 215 | C | F | 20–40 | 2 | 2 | petrous portion | 0.70956 | 0.00001 |
| | | | | | | bone pin/needle | 0.70901 | 0.00001 |
| 224 | C | F | 20–40 | 3 | 2 | petrous portion | 0.70908 | 0.00001 |
| 241a | C | F | 25–35 | 1 | 1 | petrous portion | 0.70888 | 0.00001 |
| | | | | | | femur | 0.70919 | 0.00001 |
| | | | | | | M1 root | 0.70909 | 0.00001 |
| | | | | | | M2 root | 0.70924 | 0.00001 |
| | | | | | | M3 root | 0.70922 | 0.00001 |
| | | | | | | bone pin/needle 1 | 0.70903 | 0.00001 |
| | | | | | | bone pin/needle 2 | 0.70919 | 0.00001 |
| 241b | C | U | 7–8 months | | | petrous portion | 0.70932 | 0.00001 |
| 242 | C | F? | 15–17 | 1 | 3 | petrous portion | 0.70924 | 0.00001 |
| 243 | C | F | 25–35 | 1 | 3 | petrous portion | 0.71209 | 0.00001 |
| 279 | C | F | 25–35 | 1 | 1 | petrous portion | 0.70932 | 0.00001 |
| 317 | C | M? | 20–30 | 3 | 3 | petrous portion | 0.70916 | 0.00001 |
| 350 | C | U | c. 5 | 2 | 2 | M1 enamel | 0.70940 | 0.00001 |
| 380 | C | U | c. 2 | 2 | 3 | M1 enamel | 0.70911 | 0.00001 |
| 427 | C | U | c. 6 | 2 | 3 | petrous portion | 0.70914 | 0.00001 |
| 433 | C | U | Ad | 3 | 3 | petrous portion | 0.70916 | 0.00001 |
| 439 | C | M? | 20–30 | 3 | 2 | petrous portion | 0.70910 | 0.00001 |
| 440 | C | M | 20–30 | 2 | 3 | petrous portion | 0.70929 | 0.00001 |
| 449 | C | U | c. 5 | 3 | 3 | C enamel | 0.70918 | 0.00001 |
| 459 | C | F? | 18–25 | 1 | 3 | petrous portion | 0.70916 | 0.00001 |
| 466 | C | F | 15–20 | 1 | 2 | petrous portion | 0.70911 | 0.00001 |
| 476 | I | U | 30–40 | NA | 3 | M2 enamel | 0.70946 | 0.00001 |
| | | | | | | M2 dentine | 0.70921 | 0.00001 |
| 480 | C | U | c. 6 | 2 | 2 | petrous portion | 0.70906 | 0.00001 |
| 489 | I | U | 30–40 | NA | 3 | M1 enamel | 0.70941 | 0.00001 |
| | | | | | | M1 dentine | 0.70933 | 0.00001 |
| 515 | C | F | 16–20 | 3 | 3 | petrous portion | 0.70917 | 0.00001 |
| 518 | C | M | 40+ | 3 | 2 | petrous portion | 0.70910 | 0.00001 |

Rite: C = cremation; I = inhumation. Sex: F = female; M = male; U = undetermined. Question mark indicates eventual uncertainty in the sex estimation. Absolute dates for phases and meaning of numbers for grave good complexity are explained in the paragraph "Materials". Type of sample: C = canine; M1 = first molar; M2 = second molar; M3 = third molar.

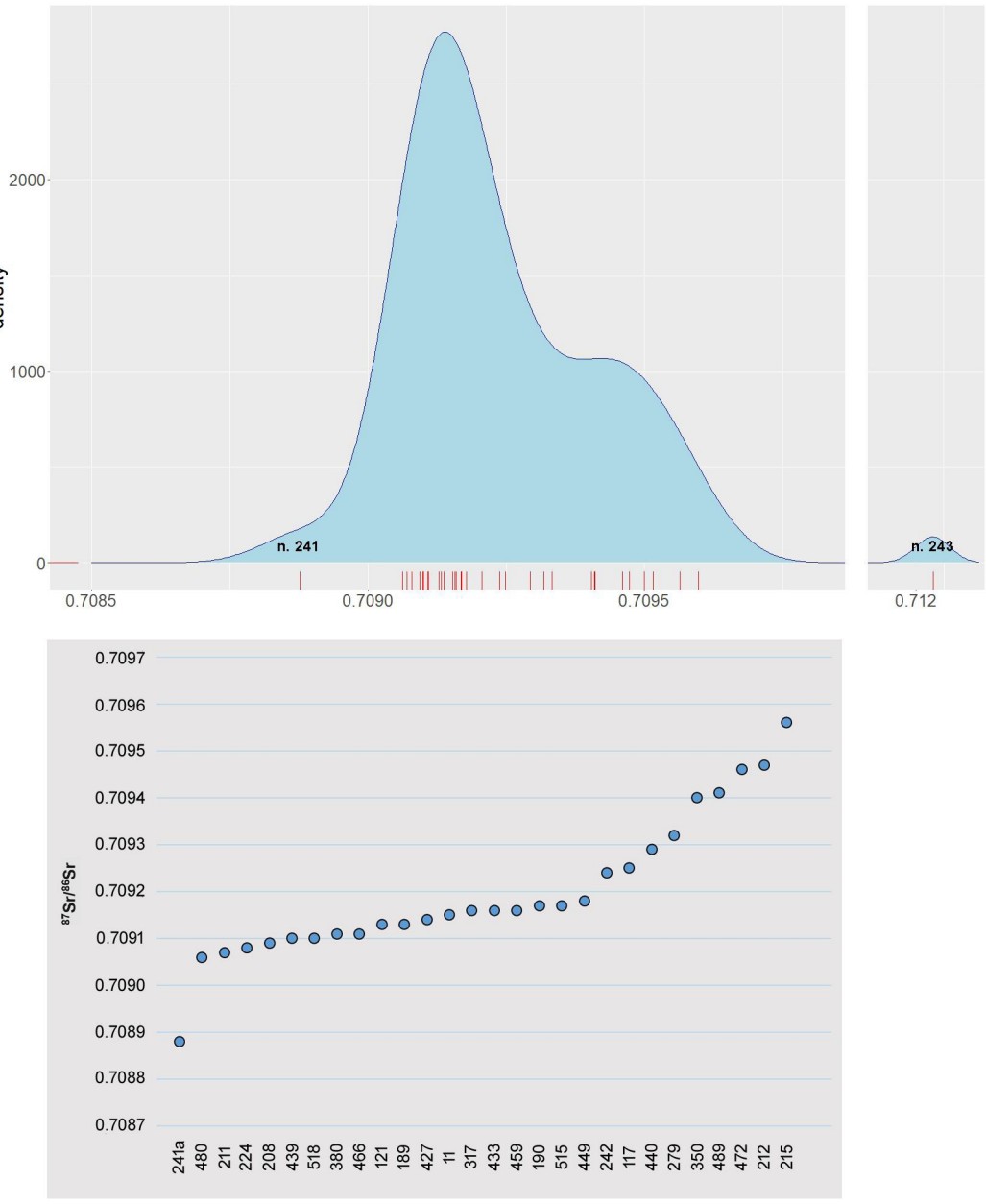

**Fig 6. Density and scatter plots.** Distribution of the $^{87}Sr/^{86}Sr$ values in the human sample. Each red bar in the density plot on the x-axis is a human sample. Outliers (n. 241 and n. 243) are highlighted.

and 0.70941 (Fig 7). Dentine isotopic ratios of these latter, however, range between 0.70907 and 0.70933. Cremations are mostly concentrated in the range 0.70906–0.70918 and, to a lesser extent, between 0.70924 and 0.70956. Two inhumations (nn. 476 and 489) fall into this second range, while only one (n. 190) is more compatible with the majority of cremations.

No significant differences have been observed between cremated and inhumated individuals, in terms of Sr isotope ratio (two tailed T-test, p = 0.75). However, removing the outliers (i.e. likely non-locals, see above), the three inhumated appears significantly different from the local cremated individuals (T-test, p = 0.016).

**Table 6. Summary statistics for 87Sr/86Sr ratios of bone and tooth specimens.**

| | |
|---|---|
| Sample size | 41 |
| MIN | 0.70888 |
| MAX | 0.71209 |
| Mean | 0.70926 |
| 2SD | 0.00094 |
| Median | 0.70917 |
| 3MADnorm | 0.00033 |
| IQR | 0.00015 |
| Tukey's fences | 0.70887–0.70948 |

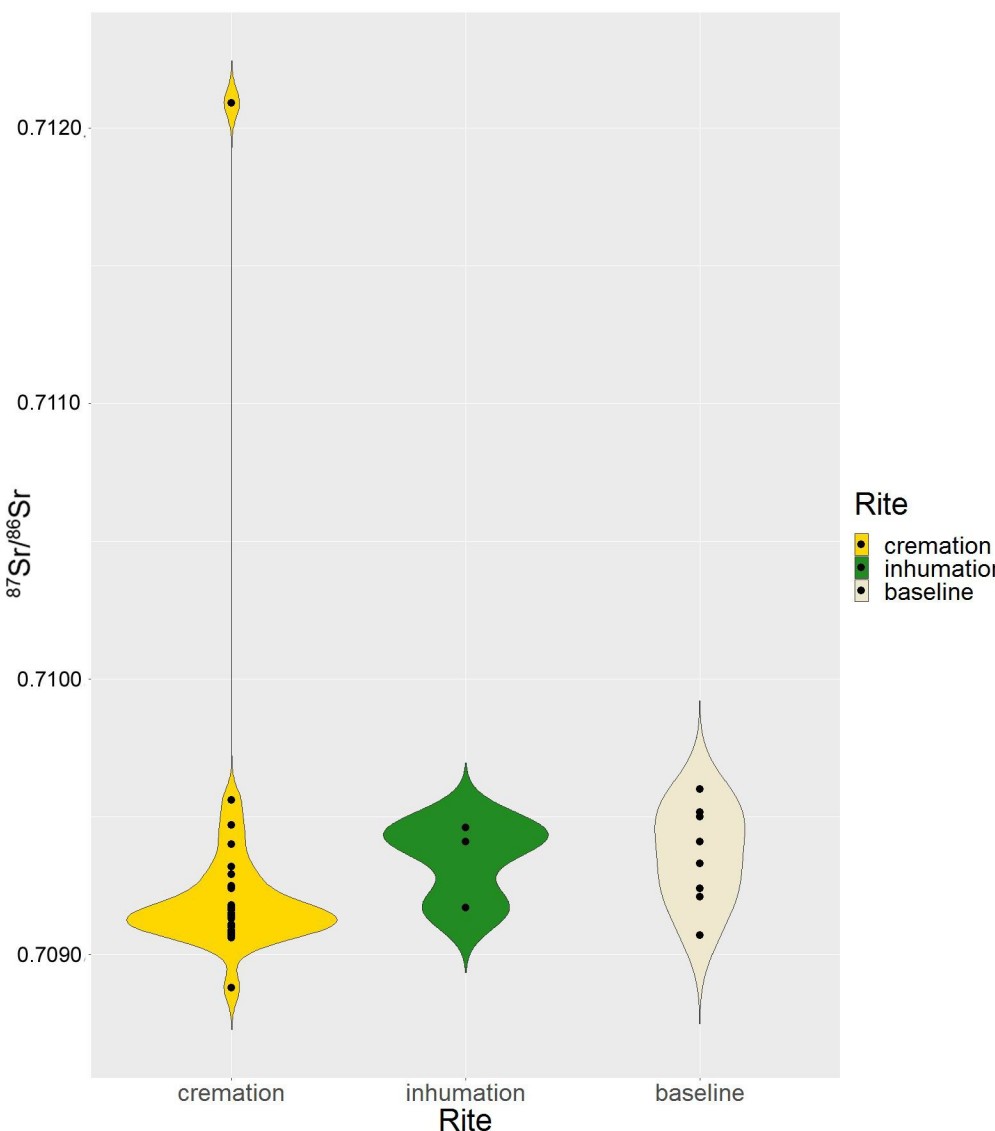

**Fig 7. Violin plot.** Distribution of the $^{87}$Sr/$^{86}$Sr values of cremations and inhumations compared with local baselines at Szigetszentmiklós.

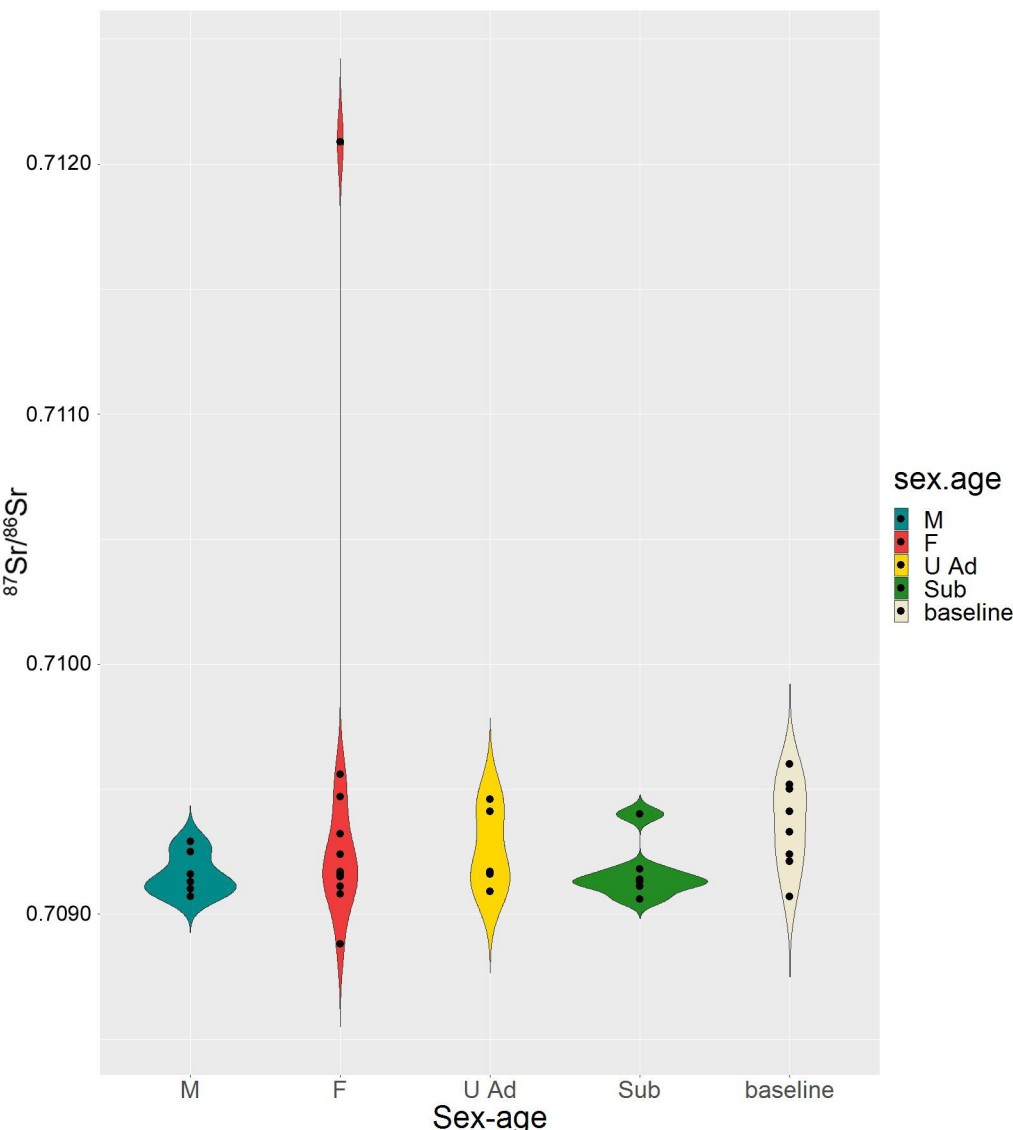

**Fig 8. Violin plot.** Distribution of the $^{87}$Sr/$^{86}$Sr values of adult males (M), females (F), adults of undetermined sex (U Ad) and subadults compared with local baselines at Szigetszentmiklós.

While values for adult females are rather variable (0.70888–0.71209), all adult males are concentrated in a narrow range (0.70907–0.70929), slightly narrower than the range for infants (0.70906–0.70940) (Fig 8). Moreover, the individuals dated to the second phase appear all compatible with local baselines (S2 Fig), and no significant differences can be observed comparing burials with different categories of grave goods (S3 Fig).

Fig 9 shows the $^{87}$Sr/$^{86}$Sr results obtained on cremated bones and dentine of the high-status adult female 241a, and on the petrous bone of one of the foetuses (241b), the two bone pins found in the urn, compared with the median of Szigetszentmiklós individuals, the mean of infants and the distribution of the local baselines. The petrous bone value for 241a (0.70888) falls outside the local range; M1 and M2 dentine, which form during the early and late childhood and partially remodel in later stages of life, appear more radiogenic and progressively more coherent with the local range. M2 and M3 dentine, the femur cortical and foetus 241b's

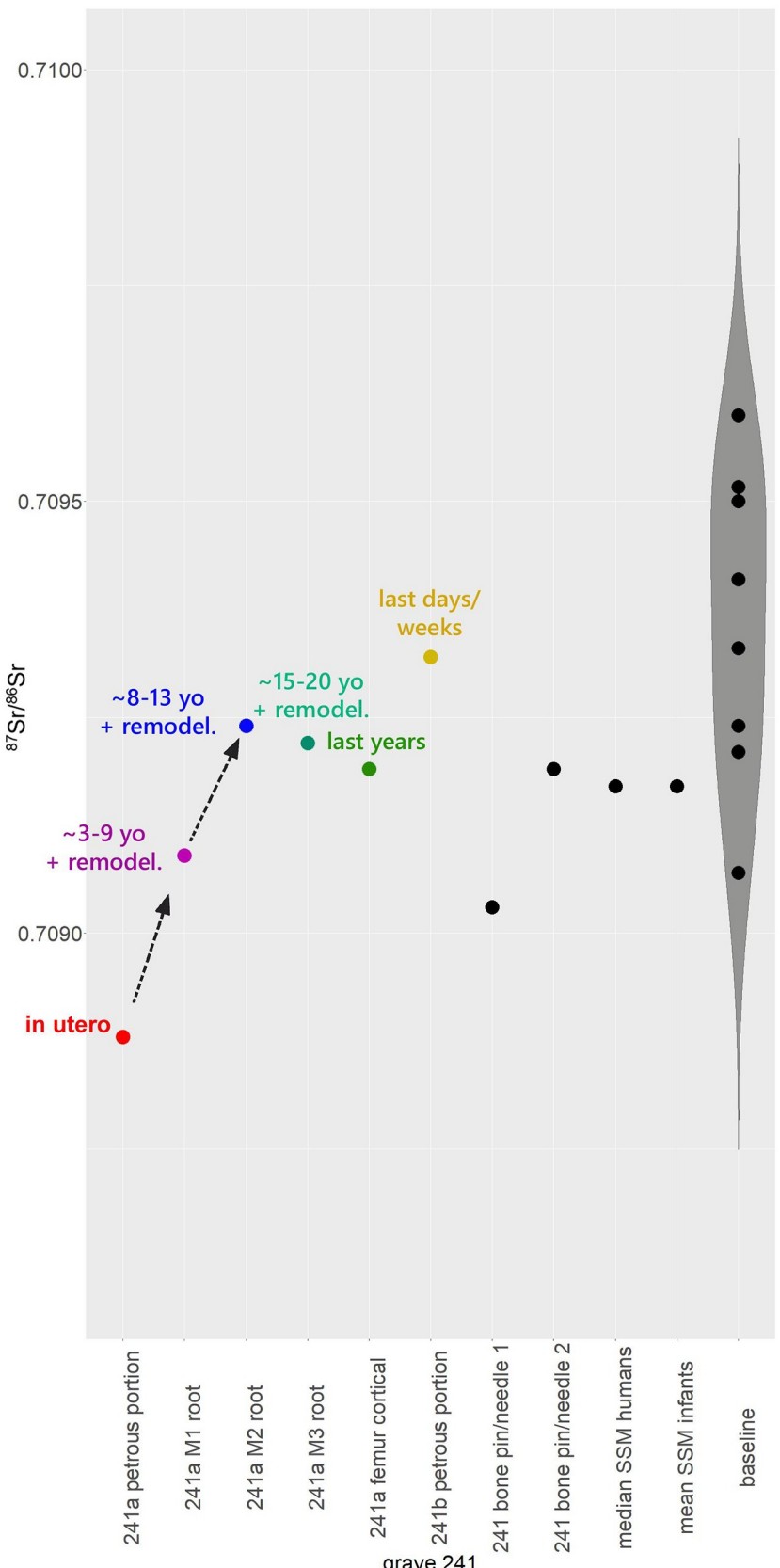

**Fig 9. Violin plot.** $^{87}$Sr/$^{86}$Sr values of various elements of the 241 burial, compared with the median of the Szigetszentmiklós individuals, the mean of infants and the local baselines.

petrous bone, which cover the period between late childhood/early adolescence and adult age prior to death, are all compatible with the local isotopic signatures. The data, therefore, suggest that the individual 241a (adult female) may have moved to Szigetszentmiklós during late childhood or early adolescence, sometime between the ages of 8 and 13. The slight increase of isotopic ratio that occurred between the remodelling of the femur cortical and the formation of 241b's petrous bone might indicate movements in the final months or weeks of her life or, more plausibly, a slight change in the diet, in terms of the nature or provenance of the ingested food, which obviously also changed seasonally among farming communities of the Bronze Age.

Concerning the two bone pins/needles, the bone material of pin/needle 2 appears local, while bone pin/needle 1 shows a less radiogenic isotopic composition, slightly outside the local range. Different provenances may be supposed, but not without doubts.

Despite the suggestive progression of the $^{87}$Sr/$^{86}$Sr values of different tissues towards the local range, as we stated above, the difference between 241a's petrous bone and local baselines is not so large (~0.0002) to assess an allochthonous provenance with certainty. For these reasons, in the next paragraph we will discuss the archaeological evidence (associated grave goods), in order to add further elements to our interpretative framework.

## Discussion

In general, most of the individuals sampled from Szigetszentmiklós-Ürgehegy appear compatible with local strontium signatures. We cannot exclude the possibility that some of these might come from a distant or even remote region, but in that is the case, this is not detectable at either an isotopic or an archaeological level. Some patterns are nonetheless revealed by the integrated analysis of archaeological, osteological, and biogeochemical data.

Despite their low number, two out of three inhumations appear isotopically slightly different from most local cremations, although still consistent with the local signatures. Interestingly, the isotope data obtained from the analysis of the inhumations found at the nearby settlements of Érd-Hosszúföldek and Százhalombatta-Belső-újföldek show that the majority of these burials fall within the range 0.7090–0.7099, which is not incompatible with the values of the area or of the broader hinterland [119]. Hence, despite the fact that the introduction of inhumation in the Vatya area around 1500 BC has been traditionally associated with the introgression of newcomers from north-west, the current isotopic data does not provide clear evidence of the presence of immigrants, at least for these specific contexts. We cannot, nonetheless, exclude the possibility that some of these individuals might have come from distant regions characterised by similar geolithologies and, therefore, strontium isotope compositions.

Our analysis has also revealed that mobility patterns vary among different categories of individuals. Infants, as expected, appear largely indigenous, but results for adult males are even more concentrated in the same narrow isotopic interval. Adult females, by contrast, show a wide variation of strontium isotope composition, although only two (n. 241 and n. 243) seem non-local. N. 243 is characterised by a highly radiogenic value, which can be found in older geolithologies, such as in the proximity of the Balaton lake [114, 115], in the Alpine valleys [49], or central Slovenia [129].

The most interesting case is the adult female buried in grave n. 241. Her urn contained two prestige objects, namely a gold hair-ring (*Noppenring*) and a bronze neck-ring (*Ösenring*), as

well as two bone pins/needles (*Knochennadeln*). This *parure*, unique among Szigetszentmiklós burials and rare among other Vatya cemeteries, clearly indicates that the woman was a member of the local elite. To judge from the appearance of her bones, her body was cremated on a large pyre, which probably burned for several hours. When the fire extinguished, the ashes were collected more carefully than usual (bone weight is 50% higher than average) and deposited in an interesting early Vatya urn. Surprisingly, the urn also contained the remains of two 7-8-month-old foetuses, which denotes that the woman was pregnant with twins, and probably died from the complications during childbirth.

The $^{87}Sr/^{86}Sr$ value obtained from her petrous bone (0.70888) falls outside the local range (0.7090–0.7097). The difference is not so large to estimate an allochthonous provenance with absolute certainty, but the overall distribution of data shows that her strontium signature diverges from the major concentration of individuals (0.70906–0.70956) (see density plot, Fig 6). Therefore, also in light of the archaeological materials, it seems plausible to hypothesize that she was not originally a member of the indigenous community and moved to Szigetszentmiklós, most likely between the ages of 8 and 13.

Among the ornaments that were part of her funeral dress, the two bone pins/needles were certainly on the pyre, since they show the typical, white-calcined appearance of cremated bone. The gold hair-ring and the bronze neck-ring do not show any clear trace of being exposed to the fire, although gold and copper/bronze melt at around 1000˚C the pyre might not have been fully reached such temperatures. More likely, they were added as grave goods during the deposition of the ashes in the urn.

Considering that the woman might have come from outside, it seems relevant to analyse the distribution of these artefacts in a broader geographical context and to explore whether there might be any compatibility between the potential provenance of the individual and of her grave goods.

The urn has been considered as one of the more significant objects that could represent some of the characteristics related to the social persona and/or personal identity of the deceased [20, 130]. The presently 30 cm-(originally probably 35–40 cm)-high urn with its 30 cm diameter belly, 10 cm-diameter bottom, long slightly curved neck with one handle and somewhat compressed ovoid body is an interesting representation of two ceramic style traditions. Its shape together with the double incised line running horizontally across the shoulder and turning downwards at a right angle on each side of the handle are typical late Nagyrév traits. Best examples are known from Szigetszentmiklós-Felsőtag, around 5 km to the East from Ürgehegy, also from Kulcs and the late Nagyrév phase of the Dunaújváros-Dunadűlő cemetery [20, 52, 57]. However, its larger size and proportions can be looked upon as a typical Vatya characteristic. It is interesting to note that this type of urns came to light from sites where the direct continuity between the Early Bronze Age Nagyrév and the Middle Bronze Age Vatya cultures could be observed. At best burial 241 can be looked upon as a representative of changing times, when this change was reflected within the changing/blending of one pottery tradition into another one. This process has been noted and described in detail earlier [20, 52, 57, 130].

## The bronze neck-ring (*Ösenring*)

The *Ösenringe* are essentially neck-rings, characterised by a high standardisation of shape, weight, and elemental composition (*Ösenringkupfer*: fahlore copper with arsenic, antimony and silver*) [131–142]. During the Early Bronze Age, in around 2000 BC, they represented both a transportable raw material, or 'ring-shaped ingots' (*Ösenringebarren)*, mainly deposited in hoards and, in the refined form usually found in high-status burials, an ornament

(*Ösenhalsringe*) (S4 Fig). The *Ösenringe* are widely distributed in Central Europe, but also occasionally documented in Scandinavia and south of the Alps, in the Po plain. However, the core regions can be identified as the Danubian region of Southern Germany and Western Austria, and nearby Bohemia, Lower Austria and Moravia, as well as in central and north-east Germany and contiguous parts of Poland, where *Ösenringe* are found almost exclusively in multityped hoards, combined with other metal objects. Interestingly, these items were made mostly of copper extracted from the Mitterberg source (Austrian Alps) [see also 134, 140, 141], which was subsequently processed in settlements occupying the lower, more fertile, and more densely inhabited river valleys of tributaries of the Danube, most notably among prosperous communities of the middle Salzach region [136].

The Hungarian plains along the Danube and Tisza rivers were included in the circulation of these items. This region is marked by the presence of *Ösenringe* in 19 hoards [21, 26, 143] and 24 burials [20, 21, 26], mostly related to female individuals. Considering the discrepancy between the isotopic composition of the petrous bone from 241a and the baselines known for the vast majority of the Great Hungarian Plain, it seems unlikely that she may have come from these Hungarian sites.

Neck-rings are not unusual in other Middle-Danubian well-known cemeteries, such as Gemeinlebarn, Franzhausen, Straubing, Singen, and other sites [e.g. 136]. They were regularly placed at the neck of the deceased, mostly for women and children from the age of 3–4 [136, 138], although, in some cases, such as at Franzhausen in Lower Austria, *Ösenringe* were also worn by men. To judge from the associated prestige objects, all these individuals were members of the local elites [144, 145].

S4 Fig also combines the density distributions of *Ösenringe* in hoards and burials: the highest density can be found in the Danubian region between Southern Bohemia, Moravia and Lower Austria, located 200–300 km to the west of Szigetszentmiklós.

## The gold ring (*Noppenring*)

The gold hair-rings *(Noppenringe* and *Lockenringe)* occur with a relative frequency in the Carpathian Basin, both in burials and in hoards, as well as in settlements, albeit much more rarely [21, 26, 146, 147] (S5 Fig). This is not surprising, as gold deposits are distributed in several districts of the Carpathian uplands, particularly in the Transylvanian Ore Mountains. Interestingly, in the mountain regions, the findings are exclusively related to hoards, while in the plains the gold hair-rings are present both in hoards and burials.

These objects are also sporadically documented outside this core region, for example, among the grave goods of the two "princely" graves of Leubingen (*Noppenringe*) and Helmsdorf (*Lockenringe*) in central Germany, dated to 1942 BC and 1840 BC, respectively [148]. This evidence reinforces the idea that these items were used to emphasise the high status of the wearer. Other gold *Noppenringe* were also found in rich graves at Franzhausen and Gemeinlenbarn [149].

## The bone pins/needles (*Knochennadeln*)

The results have shown that one of the two bone pins/needles deposed in burial n. 241 is probably locally produced (0.70919), while the other, similarly to the one found in 215, could also have been imported, as its value (0.70903) falls only slightly below the local baseline range.

This kind of object is quite common in the Early Bronze Age cemeteries of the region. At Szigetszentmiklós, they were found in 10 urns (including 241 and 215), as well as in grave n. 44 at the Vatya cemetery of Dunaújváros-Duna-dűlő. *Knochennadeln* may be dated to the Early and Middle Bronze Age type (Vatya 1 phase) and were regularly used in Early Vatya and

Maros (Szőreg-Perjámos) contexts, as well as in Nitra, Aunjetitz and Unterwölbling territories, such as at Franzhausen I and II or Gemeinlenbarn [20, 149, 150].

## Comparison with 87Sr/86Sr data from the core regions of the grave goods

Given that the *Ösenring* seems to have been imported, we sought to discover whether the regions where these artefacts are more common show local $^{87}$Sr/$^{86}$Sr baselines similar that determined for the petrous bone value (0.7088) of n.241a. Studies published to date reveal compatible ranges in Lower Austria (Franzhausen), in the Pre-Alpine Bavarian lowlands (such as the Lech Valley) and the Lake Constance area (Singen) [71, 151, 152]. Although it is impossible to assess the exact provenance of the individuals through isotope analyses, biogeochemical and archaeological evidence suggest that the woman might have come from a region along the upper course of the River Danube and wore the adornments typical for that area. However, we cannot exclude that the *Ösenring* could belong to another member of the family, of the social group, or to an ancestor.

## Conclusions

Our study demonstrates that some of the non-cremated individuals at Szigetszentmiklós-Ürgehegy appear slightly different from an isotopic perspective, compared to those who were cremated. Inhumations are nonetheless compatible with the local baselines. Further strontium/oxygen/lead isotopes and aDNA analysis might further elucidate the issue of the possible introgression of newcomers from central Europe in the Late Vatya period.

The isotopic evidence also suggests that the community of Szigetszentmiklós-Ürgehegy was largely patrilocal and practiced exogamy to a certain extent. This custom has traditionally been hypothesised and recently proven to be common across different Bronze Age societies in Europe. For example, Mittnik and collaborators have demonstrated through aDNA analysis that Early Bronze Age households of the Lech Valley "*consisted of a high-status core family and unrelated low-status individuals, a social organization accompanied by patrilocality and female exogamy*", which perpetuated over 700 years [72]. Exogamic practices have also been detected in Late Bronze Age Denmark and Northern Italy, using strontium and/or oxygen isotope analyses [45, 46, 71, 153, 154, 155].

The deceased from burial n. 241a seems to fit within this general picture and emphasises the importance of the movement of high-status female individuals. Strontium isotope analysis of her bone/tooth tissues shows that she might have moved from outside the area to Szigetszentmiklós when she was between 8 and 13 years of age, in the period of the menarche, and therefore at the beginning of the potentially fertile stage of her life. The geographic distribution of the objects which were part of her mortuary *parure*, and the *Ösenring* in particular, may indicate that she had origins in Southern Moravia, Lower Austria or in the upper Danube valley, for example in the Pre-Alpine Bavarian lowlands. All these areas show $^{87}$Sr/$^{86}$Sr baseline which are consistent with the values obtained on her petrous bone. Although archaeological and biogeochemical data converge rather significantly, we must remark that other areas of provenance could not be excluded.

Despite its distribution in more western regions, the gold hair-ring (*Noppenring*) is quite common in the Carpathian Basin. It is not improbable that the neck-ring and pins/needles were meant to symbolise a link with her native land, whereas the gold hair-ring (a wedding gift?), embodied the new local identity she acquired by joining the Szigetszentmiklós community at the highest rank. However, during adulthood, her life took a tragic turn, when she died (or was killed) whilst pregnant with (or giving birth to) twins.

On the one hand, the narrow distribution of strontium ratios among males seems to indicate patrilocality. On the other hand, the wider distribution of females, the evident allochthonous provenance of n. 243 and the 'life history' of 241a highlight the social and political role of Bronze Age women as agents of cultural hybridisation and change. Considering the increasing body of evidence about female mobility in this period, we may also argue that the integration into the kinship group of high-ranking women from outside, as a result of marriage exchanges or even rapture, might have been crucial for the emerging elite of the II millennium BC, in order to institute or reinforce political powers and military alliances, but also to secure routes, economic partnerships and, consequently, for exercising the 'redistributive power' towards the rest of the population.

## Supporting information

**S1 File. Radiocarbon dates obtained from Szigetszentmiklós inhumations and cremations.**
(DOCX)

**S1 Fig. Distribution of the total weight of bone assemblages in each category of individuals (adult males, adult females, infants).**
(TIF)

**S2 Fig. Violin plot.** Distribution of the $^{87}$Sr/$^{86}$Sr values of burials in the various phases compared with local baselines at Szigetszentmiklós.
(TIFF)

**S3 Fig. Violin plot.** Distribution of the $^{87}$Sr/$^{86}$Sr values of burials with different grave good complexity compared with local baselines at Szigetszentmiklós.
(TIFF)

**S4 Fig.** Top left: Geographical distribution of the *Ösenringe* hoards; bottom left: Density distribution of the *Ösenringe* in hoards (data gathered from [26, 134, 136, 138, 156]). Top right: Geographical distribution of the *Ösenringe* in burials; bottom right: Density distribution of the *Ösenringe* in burials (data gathered from [21, 26, 144, 148]). The maps are constructed using "Natural Earth. Free vector and raster map data @ naturalearthdata.com" available at https://www.naturalearthdata.com/downloads/10m-raster-data/
(TIF)

**S5 Fig. Geographical distribution of the gold *Noppenringe* and *Lockenringe* in burials, hoards (data gathered from [144, 146, 148]).** The map is constructed using "Natural Earth. Free vector and raster map data @ naturalearthdata.com" available at https://www.naturalearthdata.com/downloads/10m-raster-data/,.
(TIF)

## Acknowledgments

We thank Anna Cipriani for the use of geochemical facilities at University of Modena and Reggio Emilia, Kate Sharpe for her preliminary review of the manuscript, and the anonymous reviewers for their constructive suggestions. We also thank to the Ferenczy Museum Center, Szentendre, for the permissions to study the materials.

## Author Contributions

**Conceptualization:** Claudio Cavazzuti, Tamás Hajdu, Magdolna Vicze, Viktória Kiss.

**Data curation:** Claudio Cavazzuti, Federico Lugli, Magdolna Vicze.

**Formal analysis:** Claudio Cavazzuti, Federico Lugli.

**Funding acquisition:** Viktória Kiss.

**Investigation:** Claudio Cavazzuti, Magdolna Vicze, Aniko Horváth, István Major, Mihály Molnár, László Palcsu, Viktória Kiss.

**Methodology:** Claudio Cavazzuti, Federico Lugli, Alessandra Sperduti.

**Project administration:** Viktória Kiss.

**Resources:** Viktória Kiss.

**Supervision:** Claudio Cavazzuti, Tamás Hajdu, Alessandra Sperduti, Magdolna Vicze, Viktória Kiss.

**Validation:** Claudio Cavazzuti, Federico Lugli, Alessandra Sperduti, Magdolna Vicze, Viktória Kiss.

**Visualization:** Claudio Cavazzuti.

**Writing – original draft:** Claudio Cavazzuti, Federico Lugli, Alessandra Sperduti, Magdolna Vicze, Viktória Kiss.

**Writing – review & editing:** Claudio Cavazzuti, Tamás Hajdu, Alessandra Sperduti, Magdolna Vicze, Viktória Kiss.

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
