## [Decision Letter · Decision Letter 0]

25 Feb 2021

PONE-D-20-34426

Human mobility in a Bronze Age Vatya ‘urnfield’ and the dynamic life of a high-status woman

PLOS ONE

Dear Dr. Cavazzuti,

Thank you for submitting your manuscript to PLOS ONE. After careful consideration, we feel that it has merit but does not fully meet PLOS ONE’s publication criteria as it currently stands. Therefore, we invite you to submit a revised version of the manuscript that addresses the points raised during the review process.

Both reviewers recognize the interest of the work. However, they point to some problems of over-interpretation of limited isotopic changes. Reviewer 2 suggests statistical support for such interpretation. I agree with Reviewer 1 that the use of bone and dentine to establish a strontium isotopic baseline has to be considered with great caution. Please consider carefully the suggested modifications of the structure of the text and warning of potential plagiarism. Both reviewers mentioned pertinent up-to-date publications to consult. Moreover, the number of figures in the main text should be reduced, either by combining them and/or moving some of them in the supplementary data. 

We look forward to receiving your revised manuscript.

Kind regards,

Dorothée Drucker

Academic Editor

PLOS ONE

Journal Requirements:

2. In your Methods section, please provide additional information regarding the permits you obtained to collect samples for the present study. Please ensure you have included the full name of the authority that approved the field site access and, if no permits were required, a brief statement explaining why.

3. In your manuscript, please provide additional information regarding the specimens used in your study. Ensure that you have reported specimen numbers and complete repository information, including museum name and geographic location. If permits were required, please ensure that you have provided details for all permits that were obtained, including the full name of the issuing authority, and add the following statement: 'All necessary permits were obtained for the described study, which complied with all relevant regulations.' If no permits were required, please include the following statement: 'No permits were required for the described study, which complied with all relevant regulations.' For more information on PLOS ONE's requirements for palaeontology and archaeology research, see https://journals.plos.org/plosone/s/submission-guidelines#loc-paleontology-and-archaeology-research.

5. We note that Figure(s) 1, 3, 12, 13, 14 and 15  in your submission contain map images which may be copyrighted. All PLOS content is published under the Creative Commons Attribution License (CC BY 4.0), which means that the manuscript, images, and Supporting Information files will be freely available online, and any third party is permitted to access, download, copy, distribute, and use these materials in any way, even commercially, with proper attribution. For these reasons, we cannot publish previously copyrighted maps or satellite images created using proprietary data, such as Google software (Google Maps, Street View, and Earth). For more information, see our copyright guidelines: http://journals.plos.org/plosone/s/licenses-and-copyright.

a) You may seek permission from the original copyright holder of Figure(s) 1, 3, 12, 13, 14 and 15 to publish the content specifically under the CC BY 4.0 license. 

Reviewers' comments:

Reviewer's Responses to Questions

**Comments to the Author**

1. Is the manuscript technically sound, and do the data support the conclusions?

Reviewer #1: Partly

Reviewer #2: No

2. Has the statistical analysis been performed appropriately and rigorously? 

Reviewer #1: I Don't Know

Reviewer #2: No

3. Have the authors made all data underlying the findings in their manuscript fully available?

Reviewer #1: Yes

Reviewer #2: Yes

4. Is the manuscript presented in an intelligible fashion and written in standard English?

Reviewer #1: Yes

Reviewer #2: Yes

5. Review Comments to the Author

Reviewer #1: The authors present a inter-disciplinary study 3 inhumations and 26 cremations. I value this approach to compare both funerary practices using strontium isotope ratios and I really enjoyed reading the methodology used to investigate different life stages of individual 241. However, I worry that the authors over-interpreted their data as most of the discussed differences in strontium isotope ratios are around 0.0001-0.0002, which is very very small.

See below for specific comments:

L127-128 – “A total of 40 samples for 87Sr/86 127 Sr were collected for analysis from 30 burials that included 26 urn cremations and 3 inhumations” – 26 + 3 = 29, not 30?

L157 – “All burials contained the remains of a single individual” – with cremation is it very difficult (if not impossible) to be certain that we are dealing with a single individual. This sentence should be re-written to reflect this aspect of cremation research.

L230-231 – “If environmental samples are not available, or contaminated by modern sources [e.g. 86], human bone and dentine can be used as baselines” – this is not true. Bone and dentine will pick up strontium from the soil but depending on the geology this can significantly differ from the biologically available strontium for two reasons:

1- The endogenous strontium of bone and dentine might not be fully replaced by the soil strontium. Therefore, the measured strontium isotope ratio is likely to provide a value in between the original endogenous signal and that of the soil

2- It has been shown that soil strontium (bulk and leachates) have different strontium isotope ratios compared to the plants growing on these soils. The soils, therefore, do not represent adequately the biologically available strontium of the human food chain.

The sentence, as written here, will encourage bad practices as it seems that using bone and dentine is “ok”. It is not! If no adequate samples are available, and bone and dentine are indeed the only option, it needs to be clearly stated that it is not ideal.

L238-239 – “Cremated petrous portions were sampled using the method 239 reported by Jorkov et al. [87]” – Sampling petrous parts is extremely complicated, and a recent study shows the limitations of the method of Jorkov et al. I fully understand that this paper was not out yet when this study was done, but I encourage the authors to check it out, and potentially adapt their sampling method for future work: https://onlinelibrary.wiley.com/doi/abs/10.1002/rcm.9038

L298-299 – “To elude the impact of modern contamination [e.g. 86,104], we avoided sampling modern plants to build baselines for Szigetszentmiklós-Ürgehegy.” – While I fully agree that modern anthropogenic contamination could affect the strontium isotope ratio of modern plants, it does not mean they are a bad proxy, sampling just has to be done carefully. It is true, however, that the site is located in a heavily industrialised / build area and it will be difficult to find adequate samples. I would simply add, at the beginning of this sentence the following: “The area around the site is heavily affected by anthropogenic activity and, to elude…”

L329 – same comment as for L157

L410-411 – “Most of the individuals (N=17; 68.6% of the sample) are concentrated in a narrow range between 0.70908 and 0.70918. Another cluster ranges from 0.70924 to 0.70956.” When looking at such small variations, you need to consider analytical error and biological variability. Two long bone fragments from the same individual can sometimes differ from 0.0002.

L413-414 – “only 2 individuals (6.9% of the analysed sample) fall unambiguously outside the local signatures”. I totally agree that n. 243, with a value of 0.7121 clearly stand out, I have a problem with n. 241 for two reasons:

1- Only the petrous part (with a value of 0.7089) is outside the range, M1, M2 and M3 as well as the calcined femur have values > 0.7090.

2- Can we really say that 0.7089 “fall unambiguously outside the local range” with a value of 0.7089, especially seeing how the ranges were defined? Not really. I would be much more careful with this interpretation.

L421-422 – “Slight differences can be therefore be observed among individuals buried with different funerary rituals.” All 87Sr/86Sr of the (only) 3 inhumations fall within the range observed in the cremations… I don’t think we can attest any differences between the two groups based on such low numbers that all fall within a very narrow range (0.7090 – 0.7096).

L437-439 – “The data, therefore, suggest that the individual 241a (adult female) moved to Szigetszentmiklós during late childhood or early adolescence, sometime between the ages of 8 and 13.” I really like the work that has been done and the sampling strategy is really nice. However, seeing the very very small differences seen between the different skeletal elements (0.0001-0.0002), I don’t think we can make such claims. If the differences were > 0.0005 I would totally agree, but here I recommend caution. See Plomp et al. (https://onlinelibrary.wiley.com/doi/pdf/10.1002/ajpa.24059).

L440-442 – “might indicate movements in the final months or weeks of her life or, more plausibly, a slight change in the diet, in terms of the nature or provenance of the ingested food.” It is indeed likely to be diet more than anything. However, I would not describe it as a change but as a snapshot. The petrous part of the Utero is likely to reflect only a few months of the mother’s diet which could be changing seasonally, for example. This needs to be addressed more carefully, especially seeing the very very small different between M2, M3 and the femur (ca. 0.0001).

L442-443 – “The bone pins/needles, despite the similar typology, seem to be made of bone with different provenances” – I don’t agree, for the same reason as stated above. A difference of 0.0001 is not enough to say they have different provenances.

L467-507 – This discussion needs to be reviewed following the comments stated above.

L554-557 – See L442-443

L590-591 – I disagree – see L421-422

L604-608 – See L440-442

A total of 15 figures might be a bit much.

Reviewer #2: This article provides interesting Sr isotope data from cremations and inhumations from Szigetszentmiklós-Ürgehegy in Hungary. Although the data are solid, the interpretation of the data needs to be refined. The authors tend to over-interpret the data. Moreover, a statistical analysis of the data, a quantitative support of their findings, is missing, but it is needed. Small isotopic differences between individuals are easily interpreted as "mobility" or "non-local". Most parts of the article, including the abstract, would benefit from restructuring. It was challenging to understand the purpose of the study, which samples were analysed and which methodologies were used, due to the somewhat illogically structured text and incorrectly stated numbers. In addition, a large number of references are missing, appear to have been used incorrectly and some text has been taken from other articles without modification. In summary, the dataset is very interesting and of great value, partly because the database with Sr data from prehistoric cremations is still very limited. However, the article itself should be restructured and a statistical analysis of the data applied before it can be considered for publication in PLOS.

INTRODUCTION

L69. I haven’t checked these references, but are the authors sure that all these references indeed refer to ‘strontium isotope analyses to cremated materials, and verify the reliability of 87Sr/86Sr data on the petrous portion of the temporal bone’? I do miss a recent Veselka paper (Veselka et al 2021 – Rapid Communications in Mass Spectrometry).

L70. ‘We apply these methods’: be more specific. What has been applied to what?

L74-83. This section should be included in the previous section in which the authors should explain their study in more detail, or to the materials section. The ‘biographic’ approach is not new and widely applied on archaeological and historical skeletal assemblages; a specific reference in text to one appreciated researcher seems odd and not necessary.

Introduction in general: the introduction ideally ends with a paragraph in which the authors explain their aims and purposes and introduce the chosen study methodology to answer the research questions. The introduction as it is now would benefit from a major restructure. It is mentioned that “these methods” are applied to the site of Szigetszentmiklós-Ürgehegy; however, it is not explained what methods, which material, why (context) and which research questions can potentially be answered with the generated data.

THE BRONZE AGE ‘URNFIELD’ AT SZIGETSZENTMIKLÓS-ÜRGEHEGY

L120. ‘Three of those five were sufficiently preserved and selected in order to verify their

provenance’ should be moved to methods.

MATERIALS AND SAMPLING STRATEGY

L127. If I add up all samples taken, more than 40 samples have been analysed.

3 burnt enamel infant

23 burnt PP

M1 burials 121 and 380 = 2 burnt enamel

3 * 2 = 6 enamel inhumation

3 * 2 = 6 dentine inhumation

M123 = 3 dentine B241

Femur= 1 bone B241

2 * PP foetus = 2 PP foetus B241

2 * needle B241

Is a total of 48 samples??

L133. Dentine is very susceptible for diagenetic alterations (e,g. Budd et al, 2000). The authors should either explain the reason for sampling the dentine and refer to the diagenesis problematics. If the dentine samples are taken for baseline/reference purposes, the authors must reflect on the fact that they are assuming 100% diagenetically altered dentine. After all, if biogenic Sr ratio of the dentine was 0.712, the diagenetically altered Sr ratio could be somewhere between 0.712 and the local Sr ratio, thus also exhibit a ratio of 0.710.

L136. 26 cremation – 3 infants = remaining 23 burials. Not 24.

L137-138. ‘selecting those individuals with both preserved’. What do the authors mean? Both PP preserved? If so, why? Why would you only select those individuals that have a left and a right PP preserved in their cremated remains?

L143-144. I disagree with the fact that the authors find it ‘reasonable to assume’ that the mothers did not migrate during the first 2 years of the child’s life. Archaeological and historical Sr data on dental enamel show childhood mobility in (pre)historic populations. It is a dangerous assumption to be made, but one that the whole concept of Sr isotope analysis in archaeology is based upon. However, it is never ‘reasonable to assume…’ if you don’t have archaeological or historical data to verify that assumption.

L143-144. This sentence is taken from Cavazzuti et al. 2019. I noticed quite some sentences that have been copy pasted or minimally modified from other papers…. (plagiarism: carefully check the manuscript!).

L149. Reference 68 does not refer to PP research. The authors should carefully check all references and use primary references instead.

L149. The Veselka et al. paper should be mentioned here as well.

L153. The data is Ref 45 do not show the reliability of the PP, as the PP data are compared to that of the M2 that mineralises after the PP stops remodelling.

L170. 2nd dentine is laid down in the pulp chamber. The authors sampled the outer surface of the root. And only a few mg. How can 2nd dentine be of interest for this study?

L179. There are many more factors contributing to the turnover rate of bones (biological sex, health, age, location within the bone, stress, etc.). It is therefore almost impossible to predict the turnover rate of the fragment of bone selected for this study. Moreover, IF the average bone remodelling rate is estimated to be 3-8 years, the femur does not reflect the last few year of life, but the last 10 to 30 years (i.e., decades) of life.

Materials in general: the text should be restructured, and the authors should systematically refer to all materials sampled. The text as in its current form is a hybrid text with sampled material and background information.

METHODS: SR ISOTOPE ANALYSIS

L217-226. No references are provided.

L231. “human bone and dentine can be used as baseline”. This approach is based on the very dangerous assumption that the bone and dentine samples are 100% diagenetically alterated. This text requires additional information.

L238-241. This text is copy pasted from Cavazzuti et al., 2019 (plagiarism). Rewrite.

L234. This technique (sr isotopes) has been applied in archaeology for 30+ years. The technique itself is much older.

L247. 30 ul columns? This seems incorrect to me.

L259. Please provide you blank data.

METHODS: GEOLOGY OF CENTRAL HUNGARY AREA AND BIOLOGICALLY AVAILABLE STRONTIUM ‘ISOSCAPE’

L307. If I understand correctly, results are given in this section? Results should be presented in the results section. I would include a paragrapgh about the additional baseline dat ain the results section, strengthening the range you presented in this section.

RESULTS OF THE SR ANALYSIS

L414-415. Please provide the data for both females.

L427. You literally have one statistical outlier in phase 1. This is no convincing evidence of a “progressive reduction of mobility throughout the phases”, but overinterpretation of the data.

L433. 1st Dentin is unable to remodel, it never remodels. Only 2nd and 3rd dentine are formed/added during life (and secondary dentin does remodel), but the primary dentine does not remodel. The samples taken for this study are not selected from the pulp cavity, hence the data ate probably not influenced at all by secondary dentine.

L436. No, not the foetus PP which you also refer to.

L439. Be cautious. The isotopic differences between the samples are very very small and intra-individual isotopic differences do occur. You should avoid overinterpreting the data.

L 44. Same here. The difference between the pins extremely limited, also taken the ±2SE into account. These data do not necessarily point toward 2 different regions of origin.

Results in general: the authors tend to overinterpret the data. Moreover, the one datapoint of interest, the 0.7129 is not identified (male/female. Etc etc). Moreover, a statistical analysis of the data is absent that would probably underline the homogeneity of the data (except one). The interpretation of the possible non-local origin of 241 (0.7088) could also be strengthened using statistics.

DISCUSSION

L467. No, the data are consistent with local, but the individuals could still be of a non-local origin.

L468-470. Indeed, so don’t state that they “appear largely local”

L478. Reference?

L480. In this study only 3 (!!) inhumed individuals are included. One cannot and should never ‘weaken a hypothesis’ based on only 3 datapoints. Try not to overinterpret the data,

L486. Now the outlier is identified in the text, 243. I agree that 243 is of non-local descent. The data for 241, however, nis not convincing.

L498. The authors should first check the statistically significant. Without this data, the interpretation in L498 seems incorrect.

L509-563

Now the artefacts are discussed in great detail. This part of the research is, however, not properly introduced in the introduction. Should it indeed be included in the text, or transferred to the supplementary data? I do see the link between the artefacts and 241, but I would suggest to restructure the text completely, and focus on 241 and her grave goods in a dedicated paragraph.

L575. Be careful not to overinterpret. Maybe the ornaments have been in the family for decades and do not reflect the female’s origin, but that of her tribe/culture/family. This complexity of prehistoric populations and their social interactions and family links cannot be easily solved and alternative interpretations should be reflected upon as well.

CONCLUSIONS

L590. Overinterpretation of the data. Please provide statistical evidence.

L594. aDNA on well cremated remains?

L595 - end. Again, suggestive. The data are not strong enough to provide solid evidence for this conclusion.

Minor (textual) changes

Abstract

L29. The comma (,) should be a semi-colon (;)

L32. Use the oxford comma after ‘apogee’

L37. Here “our results” are presented, but ideally in an abstract, the applied methodology should be mentioned as well. This is absent in the abstract and should be added.

L41-43. The abstract should mention the results, a synopsis of the work executed. This section mentioned the work (Sr analysis) that have been done, but fails to mention the (interpret) data.

L328. Comma before ‘with’.

L332. Old= of age

L474. Isotopic= isotope

6. PLOS authors have the option to publish the peer review history of their article (what does this mean?). If published, this will include your full peer review and any attached files.

Reviewer #1: No

Reviewer #2: No

---

## [Author Response · Author response to Decision Letter 0]

21 Apr 2021

Dear Editor and Reviewers,

we are very grateful to you for your work in reviewing our manuscript and for providing comments, which we consider extremely valuable in the process of improving the overall quality of the paper. We also thank you for recognizing its merit.

General points:

As you can see below and in the revised manuscript, we restructured and changed several parts of the manuscript following your suggestions. We agreed with the idea of interpreting the results with more caution, also supporting the strontium data with statistical support (see final part of the “Methods: strontium isotope analysis”, and “Results” – Table 7). We also changed the title, the abstract and the discussion/conclusion, accordingly. We acknowledge that the high-status woman 241a is characterized by a strontium value which is, albeit outside the local range, not so distant from the less radiogenic baselines. Moreover, she appears also separated by the main concentration of the individuals’ distribution. It was exactly for this ambiguous position, at the edge between local and non-local signatures, that we engaged in a more articulated analysis of her different tissues, of the grave good geographical distribution, as well as in the strontium isotope analysis of the two needles deposed in the urn. Of course, the interpretation about her allochthonous provenance can only be hypothetical, and no specific origin can be assessed with precision. Therefore, we changed the text in order to emphasize that our attempt was mainly to use multiple sources and perspectives in the analytical process, an approach that is not original, but also not so frequently employed.

Concerning the use of bone/dentine for strontium baselines, we modified the text, emphasizing that these tissues have been used only to support published data from vegetal, waters, archaeological fauna from the Budapest hinterland. Unfortunately, we could not take environmental samples, because the area is heavily urbanized. We could not even use faunal remains, as the no animal bones were found in the cemetery area, and the associated settlement has never been discovered. However, we mentioned Giblin et al.’s work (2019; ref. 106), which reports 17 strontium data obtained from faunal remains found at Érd-Hosszúföldek Bronze Age site, which is contemporary to SSM and is located around 10 km to the west on the opposite bank of the Danube (same alluvial basin).

We also change the few sentences in Methods section which were indeed used in another paper, written by me (Claudio Cavazzuti) as main author a couple of years ago and that was published on the same journal, PLOS ONE. Anyway, we reformulated the sentence in order to avoid plagiarism. 

References suggested by the Reviewers were added, or changed when incorrect, and the number of figures was reduced.

As you can see in the resubmission, we also added radiocarbon dates obtained on 9 cremations and 2 inhumations in the Supplementary Materials and we also included the people who performed the analysis at the ICER Centre, Institute for Nuclear Research, Debrecen (HUN). While the data from bone collagen (inhumations) seem reliable, those on cremated tissues show a complete incompatibility with the relative chronology, reconstructed with an accurate analysis of urn typology and grave goods, which is a well-established and rather precise method for dating Bronze Age materials in Hungary, as well as in most part of Europe. Just to make an example, all the materials (grave goods and vessels) of the 241 point to 2000/1900 BC, while the 14C indicates 1740-1500 cal. BC (2 sigma). Such a difference of minimum 200 years does not seem explainable with the “old wood effect”, so we preferred to report and critically discuss the data in the SM. Even more unreliable appears 459’s date, which ranges between 1390-1050 cal. BC (2 sigma), at least 500 years younger than the expected phase.

In the following pages, we responded to the Reviewers’ comments in detail, and we again thank you for the time and attention given to our work.

Reviewer #1: The authors present a inter-disciplinary study 3 inhumations and 26 cremations. I value this approach to compare both funerary practices using strontium isotope ratios and I really enjoyed reading the methodology used to investigate different life stages of individual 241. However, I worry that the authors over-interpreted their data as most of the discussed differences in strontium isotope ratios are around 0.0001-0.0002, which is very very small.

See below for specific comments:

L127-128 – “A total of 40 samples for 87Sr/86 127 Sr were collected for analysis from 30 burials that included 26 urn cremations and 3 inhumations” – 26 + 3 = 29, not 30?

Yes, thanks. We rephrase as follows: A total of 40 samples for 87Sr/86Sr were collected for analysis from 30 individuals from 29 burials (3 inhumations and 26 urn cremation; n. 241 includes three individuals, two of them analysed) (Table 1).

L157 – “All burials contained the remains of a single individual” – with cremation is it very difficult (if not impossible) to be certain that we are dealing with a single individual. This sentence should be re-written to reflect this aspect of cremation research.

We agree with the claim for more caution and rephrase as follows: “Except for burial n. 241, which included the cremated remains of an adult female (n. 241a) and of two foetuses of 28-32 gestational weeks (nn. 241b, 241c), all the other burials seemed to contain the remains of a single individual, since no exceeding bone was found in the assemblages. Moreover, the total weight of bone assemblages was always lower than the expectations for a complete cremation of a single individual reported in literature [61–66].”

L230-231 – “If environmental samples are not available, or contaminated by modern sources [e.g. 86], human bone and dentine can be used as baselines” – this is not true. Bone and dentine will pick up strontium from the soil but depending on the geology this can significantly differ from the biologically available strontium for two reasons:

1- The endogenous strontium of bone and dentine might not be fully replaced by the soil strontium. Therefore, the measured strontium isotope ratio is likely to provide a value in between the original endogenous signal and that of the soil

2- It has been shown that soil strontium (bulk and leachates) have different strontium isotope ratios compared to the plants growing on these soils. The soils, therefore, do not represent adequately the biologically available strontium of the human food chain.

The sentence, as written here, will encourage bad practices as it seems that using bone and dentine is “ok”. It is not! If no adequate samples are available, and bone and dentine are indeed the only option, it needs to be clearly stated that it is not ideal.

Thanks for comment. We rephrase as follows: “The possible provenance of individuals is estimated by comparing the ratio between strontium-87 and strontium-86 in bones/teeth with the local baseline values measured in faunal/vegetal samples, as well as soils and waters, from the burial site or its geologically coherent immediate hinterland. If environmental samples are not available, or contaminated by modern sources [e.g. 86], archaeological fauna and human bone/dentine can be used to support local baselines, although, especially the latter, is not an ideal option and must be considered with caution”.

L238-239 – “Cremated petrous portions were sampled using the method 239 reported by Jorkov et al. [87]” – Sampling petrous parts is extremely complicated, and a recent study shows the limitations of the method of Jorkov et al. I fully understand that this paper was not out yet when this study was done, but I encourage the authors to check it out, and potentially adapt their sampling method for future work: https://onlinelibrary.wiley.com/doi/abs/10.1002/rcm.9038

Unfortunately, we read this important methodological contribution only after we carried out the analysis and submitted the manuscript. We will certainly apply this new method in future work.

L298-299 – “To elude the impact of modern contamination [e.g. 86,104], we avoided sampling modern plants to build baselines for Szigetszentmiklós-Ürgehegy.” – While I fully agree that modern anthropogenic contamination could affect the strontium isotope ratio of modern plants, it does not mean they are a bad proxy, sampling just has to be done carefully. It is true, however, that the site is located in a heavily industrialised / build area and it will be difficult to find adequate samples. I would simply add, at the beginning of this sentence the following: “The area around the site is heavily affected by anthropogenic activity and, to elude…”

Thanks for the comment. We rephrase as follows: “The few cultivated fields are very close to houses and factories and have been heavily manured in the recent past, as well as today. The area around the site is heavily affected by anthropogenic activity and, to elude the impact of modern contamination [e.g. 86,104], we avoided sampling modern plants to build baselines for Szigetszentmiklós-Ürgehegy.”

L329 – same comment as for L157

We rephrased as follows: “The cremated bones of the 26 analysed burials showed a rather homogenous, calcined aspect, which indicated a relatively high temperature reached by the pyre (>800°C) [28,106–110]. With the exception of n. 241, all the other 25 urns seemed to contain the remains of a single individual, since no exceeding bone was found in the assemblages. Moreover, the total weight of bone assemblages was always lower than the expectations for a complete cremation of a single individual reported in literature [61–66].”

L410-411 – “Most of the individuals (N=17; 68.6% of the sample) are concentrated in a narrow range between 0.70908 and 0.70918. Another cluster ranges from 0.70924 to 0.70956.” When looking at such small variations, you need to consider analytical error and biological variability. Two long bone fragments from the same individual can sometimes differ from 0.0002.

We agree the difference is small, indeed, if we look at absolute numbers. However, relatively to the great geolithological homogeneity of the alluvial plain, and to the available strontium baselines, which all range between 0.7090 and 0.7097, we need to take into consideration the hypothesis of small but meaningful variations. Concerning the analytical error, even propagated with the external reproducibility of the reference materials, it is on the 5th decimal, thus almost negligible for our considerations.

L413-414 – “only 2 individuals (6.9% of the analysed sample) fall unambiguously outside the local signatures”. I totally agree that n. 243, with a value of 0.7121 clearly stand out, I have a problem with n. 241 for two reasons:

1- Only the petrous part (with a value of 0.7089) is outside the range, M1, M2 and M3 as well as the calcined femur have values > 0.7090.

2- Can we really say that 0.7089 “fall unambiguously outside the local range” with a value of 0.7089, especially seeing how the ranges were defined? Not really. I would be much more careful with this interpretation.

We agree with being more cautious and we rephrased as follows: “Considering the estimated isotopic ranges for Szigetszentmiklós (0.7091-0.7095) and those for its hinterland (0.7090-0.7097), only 1 individual, namely adult female n. 243, falls unambiguously outside the local signatures. Burial n. 241 of a rich adult female shows a slightly less radiogenic value (0.70888), which falls outside the local range. Hypothetically n. 241 might be considered non-indigenous, but such interpretation needs further analyses, both from a biogeochemical and archaeological perspective. In fact, while n. 243 does not contain any grave goods at all, n. 241 is equipped with a set of prestigious items, which may add further elements to the interpretation. Moreover, we sampled other dental/skeletal tissues from n. 241, in order to analyse 87Sr/86Sr variations during her various life stages”.

We also performed a statistical analysis on the data in order to apply some caution in the following interpretation. Therefore, we added a paragraph in the Methods and Results sections.

Methods: “To statistically detect possible non-local individuals, we employed two approaches: the median absolute deviation from the median (MAD) and the Tukey’s fences method. Concerning the MAD, we calculated the 3MADnorm as reported in Lightfoot and O’Connell (2016) and Leys et al. (2013), namely multiplying by 3 the MAD and scaling it to a b value. This latter is usually assumed as a constant (1.4826) for normal distributions, neglecting outlier-induced abnormalities (Leys et al., 2013). Excluding outliers though a Grubb test (p < 0.05; 0.71209 is an outlier), our data appears as normally distributed (Shapiro-Wilk test; W = 0.954; p = 0.105), hence a b value = 1.4826 has been used as scaling factor for the MAD. Tukey’s interval has been calculated as Q1-1.5(IQR) and Q3-1.5(IQR), where IQR is the interquartile range (Tukey, 1977). A two-tailed T-test was also performed searching for group differences between cremated vs. inhumated individuals. We acknowledged that the small number of inhumated individuals (n = 3) may impact the prediction power of the test. All the tests have been performed manually using either Microsoft Excel or MATLAB.”

Results: “Statistical results are summarized in Table 7. Both employing 3MADnorm or Tukey’s IQR [88–90], only two individuals are detected as outliers, namely n. 215 (87Sr/86Sr =0.70956) and n. 243, the most radiogenic of the dataset (87Sr/86Sr=0.71209). N. 215, however, fall within the local strontium signatures (Szigetszentmiklós=0.7091-0.7095; immediate hinterland=0.7090-0.7097). Individual n. 241a (the lowest radiogenic; 87Sr/86Sr=0.70888) is outside the local range but lays at the lower edges (but still within) of the statistical intervals (3MADnorm or Tukey’s IQR).

Table 1. Summary statistics for 87Sr/86Sr ratios of bone and tooth specimens.

Sample size 41

MIN 0.70888

MAX 0.71209

Mean 0.70926

2SD 0.00094

Median 0.70917

3MADnorm 0.00033

IQR 0.00015

Tukey’s fences 0.70887-0.70948

No significant differences have been observed between cremated and inhumated individuals, in terms of Sr isotope ratio (two tailed T-test, p = 0.75). However, removing the two detected outliers (i.e. likely non-locals, see above), the three inhumated appears significantly different from the local cremated individuals (T-test, p = 0.016).“

L421-422 – “Slight differences can be therefore be observed among individuals buried with different funerary rituals.” All 87Sr/86Sr of the (only) 3 inhumations fall within the range observed in the cremations… I don’t think we can attest any differences between the two groups based on such low numbers that all fall within a very narrow range (0.7090 – 0.7096).

We deleted the sentence: “Slight differences can therefore be observed among individuals buried with different funerary rituals.” See also previous comment/reply.

L437-439 – “The data, therefore, suggest that the individual 241a (adult female) moved to Szigetszentmiklós during late childhood or early adolescence, sometime between the ages of 8 and 13.” I really like the work that has been done and the sampling strategy is really nice. However, seeing the very very small differences seen between the different skeletal elements (0.0001-0.0002), I don’t think we can make such claims. If the differences were > 0.0005 I would totally agree, but here I recommend caution. See Plomp et al. (https://onlinelibrary.wiley.com/doi/pdf/10.1002/ajpa.24059).

Thanks for the comment and for the reference provided. We changed the text as follows: The data, therefore, suggest that the individual 241a (adult female) may have moved to Szigetszentmiklós during late childhood or early adolescence, sometime between the ages of 8 and 13. The slight increase of isotopic ratio that occurred between the remodelling of the femur cortical and the formation of 241b’s petrous bone might indicate movements in the final months or weeks of her life or, more plausibly, a slight change in the diet, in terms of the nature or provenance of the ingested food, which obviously also changed seasonally among farming communities of the Bronze Age.

Concerning the two bone needles, the bone material of pin/needle 2 appears local, while bone/pin needle 1 shows a less radiogenic isotopic composition, slightly outside the local range. Different provenances may be supposed, but not without doubts. 

Despite the suggestive progression of the 87Sr/86Sr values of different tissues towards the local range, as we stated above, the difference between 241a’s petrous bone and local baselines is not so large (~0.0002) to assess an allochthonous provenance with certainty. For these reasons, in the next paragraph we will discuss the archaeological evidence (associated grave goods), in order to add further elements to our interpretative framework”.

L440-442 – “might indicate movements in the final months or weeks of her life or, more plausibly, a slight change in the diet, in terms of the nature or provenance of the ingested food.” It is indeed likely to be diet more than anything. However, I would not describe it as a change but as a snapshot. The petrous part of the Utero is likely to reflect only a few months of the mother’s diet which could be changing seasonally, for example. This needs to be addressed more carefully, especially seeing the very very small different between M2, M3 and the femur (ca. 0.0001).

Agree with the idea of significant seasonal change of diet among farming communities. See previous comment. We revised all that part.

L442-443 – “The bone pins/needles, despite the similar typology, seem to be made of bone with different provenances” – I don’t agree, for the same reason as stated above. A difference of 0.0001 is not enough to say they have different provenances.

We applied the caution requested by the reviewer also to the pin’s provenance. See text of the previous comment.

L467-507 – This discussion needs to be reviewed following the comments stated above.

L554-557 – See L442-443

L590-591 – I disagree – see L421-422

L604-608 – See L440-442

We revised all these parts according to the previous comments.

A total of 15 figures might be a bit much.

We reduced the number of figures.

Reviewer #2: This article provides interesting Sr isotope data from cremations and inhumations from Szigetszentmiklós-Ürgehegy in Hungary. Although the data are solid, the interpretation of the data needs to be refined. The authors tend to over-interpret the data. Moreover, a statistical analysis of the data, a quantitative support of their findings, is missing, but it is needed. Small isotopic differences between individuals are easily interpreted as "mobility" or "non-local". Most parts of the article, including the abstract, would benefit from restructuring. It was challenging to understand the purpose of the study, which samples were analysed and which methodologies were used, due to the somewhat illogically structured text and incorrectly stated numbers. In addition, a large number of references are missing, appear to have been used incorrectly and some text has been taken from other articles without modification. In summary, the dataset is very interesting and of great value, partly because the database with Sr data from prehistoric cremations is still very limited. However, the article itself should be restructured and a statistical analysis of the data applied before it can be considered for publication in PLOS.

INTRODUCTION

L69. I haven’t checked these references, but are the authors sure that all these references indeed refer to ‘strontium isotope analyses to cremated materials, and verify the reliability of 87Sr/86Sr data on the petrous portion of the temporal bone’? I do miss a recent Veselka paper (Veselka et al 2021 – Rapid Communications in Mass Spectrometry).

True. We removed the references related to C14 on cremated bones and we added Veselka et al. 2021, which was not published yet when we submitted the manuscript to PLOS ONE.

L70. ‘We apply these methods’: be more specific. What has been applied to what?

We rephrased as follows to clarify: “In our study, we apply these sampling methods and sampling strategies to the site of Szigetszentmiklós-Ürgehegy, one of the most important cemeteries of the Late Nagyrév and Vatya culture”.

L74-83. This section should be included in the previous section in which the authors should explain their study in more detail, or to the materials section. The ‘biographic’ approach is not new and widely applied on archaeological and historical skeletal assemblages; a specific reference in text to one appreciated researcher seems odd and not necessary.

We moved these lines to the “Materials” section and rephrased as follows: “Except for burial n. 241, which included the cremated remains of an adult female (n. 241a) and of two foetuses of 28-32 gestational weeks (nn. 241b, 241c), all the other burials seemed to contain the remains of a single individual, since no exceeding bone was found in the assemblage. 

Burial n. 241 is of a particular relevance, as the urn also contains a golden ring (Noppenring), a bronze neck-ring with flat-hammered ends (Ösenring, or Ösenhalsring), and two ornamental bone needles (Knochennadeln), which undoubtedly mark the individual’s high-status. 

The ‘life history’ of n. 241 assumes even more importance when re-incorporated into the wider ‘prosopography’ of Szigetszentmiklós-Ürgehegy individuals which, in turn, can be integrated into the broader, panoramic analysis of the mobility of people in the European Bronze Age [e.g. 45,46,53–56].

For this exceptional case, we applied a ‘biographic’ approach, collecting different dental and skeletal samples forming at different stages of the woman’s life (Table 2: 1-5), in order to reconstruct her movements from childhood to the pre-mortem period: the petrous bone, the dentine from M1, M2, and M3, and a cortical portion of the femur.

Introduction in general: the introduction ideally ends with a paragraph in which the authors explain their aims and purposes and introduce the chosen study methodology to answer the research questions. The introduction as it is now would benefit from a major restructure. It is mentioned that “these methods” are applied to the site of Szigetszentmiklós-Ürgehegy; however, it is not explained what methods, which material, why (context) and which research questions can potentially be answered with the generated data.

We tried to clarify as requested: “In our study, we apply these new osteological methods and sampling strategies for the analysis of strontium isotopes to a sample of 29 burials from Szigetszentmiklós-Ürgehegy, one of the most important and best-preserved cemeteries of the Late Nagyrév and Vatya culture. Located a few kilometres south of Budapest, in the northern part of the Csepel Island, the cemetery includes over 500 urn cremations and eight inhumations [50].

Our main goals are a) contributing to the understanding of cremation practices of the Early and Middle Bronze Age in Hungary; b) exploring mobility patterns among inhumations and cremations, as well among individuals of both sexes and different age class, through strontium isotope analysis; c) applying a ‘biographic’ approach, namely targeting different human tissues for strontium isotope analysis, to one particularly relevant case, that of burial n. 241, which contains multiple individuals and a variety of prestige grave goods.”

THE BRONZE AGE ‘URNFIELD’ AT SZIGETSZENTMIKLÓS-ÜRGEHEGY

L120. ‘Three of those five were sufficiently preserved and selected in order to verify their

provenance’ should be moved to methods.

We did it and rephrase as follows: “A total of 41 samples from different human tissues of 30 individuals included in 29 burials (3 inhumations and 26 urn cremations; burial n. 241 includes three individuals, two of them analysed), as well as from 3 bone pin/needles from burials n. 215 and 241a (Table 1).

Concerning inhumations, only three individuals had teeth sufficiently preserved to sample enamel and dentine for strontium isotope analysis. We sampled both dental tissues from first or second molars (M1, M2), the crowns of which develop at the age of 0-3 years and 3-8 years, respectively [54,55]. It is well known that while enamel appears, in most cases, to be a reliable reservoir of biogenic strontium, dentine is affected by diagenesis and contamination from the burial environment [56]. In fact, the original strontium isotope composition of dentine may vary post mortem up to 100% during the biochemical exchange with the soil. Therefore, although we cannot measure the exact proportion of variation, dentine’s values may be useful in a comparison to available strontium baselines for the area of the site. 

Regarding cremations, the sampling strategy was carried out according to a set of criteria to ensure future analyses on the same contralateral anatomical element, and taking into account different archaeological and biological variables, namely sex and age-at-death of the individuals, burial rite (inhumation or cremation), quality/quantity of grave goods, and chronological phase of the burials.

From the 26 cremations, we sampled fragments of non-erupted tooth enamel in three cases (infant burials nn. 350, 380, 449) and for the remaining 24 burials we used a fragment of the otic capsule of the pars petrosa (petrous portion) of the temporal bone, selecting those individuals with both preserved. The petrous portion begins to form in utero at approximately 16–18 gestational weeks, is fully ossified by the time of birth, and does not undergo any subsequent remodelling after the age of two years [57–59].”

MATERIALS AND SAMPLING STRATEGY

L127. If I add up all samples taken, more than 40 samples have been analysed.

3 burnt enamel infant → 3 (350, 380, 449)

23 burnt PP → 24 (11, 117, 121, 189, 208, 211, 212, 215, 224, 241a, 241b, 242, 243, 279, 317, 427, 433, 439, 440, 459, 466, 480, 515, 518) 

M1 burials 121 and 380 = 2 burnt enamel → only 1, namely M1 enamel of 121, was missing from the table. Added. (380 was counted two lines above)

3 * 2 = 6 enamel inhumation → only 3 (burials 190, 476, 489)

3 * 2 = 6 dentine inhumation → only 3 (burials 190, 476, 489)

M123 = 3 dentine B241 → 3

Femur= 1 bone B241 → 1

2 * PP foetus = 2 PP foetus B241 → only 1 (of the foetus 241b), already counted few lines above among the 24 PP)

2 * needle B241 → 2 from 241 + 1 from burial n. 215

Is a total of 48 samples??

The recounted total is therefore 41 (121’s M1 enamel was missing) from the table.

L133. Dentine is very susceptible for diagenetic alterations (e,g. Budd et al, 2000). The authors should either explain the reason for sampling the dentine and refer to the diagenesis problematics. If the dentine samples are taken for baseline/reference purposes, the authors must reflect on the fact that they are assuming 100% diagenetically altered dentine. After all, if biogenic Sr ratio of the dentine was 0.712, the diagenetically altered Sr ratio could be somewhere between 0.712 and the local Sr ratio, thus also exhibit a ratio of 0.710.

Correct. We rephrased as follows, adding Budd’s et al reference: “It is well known that while enamel appears, in most cases, to be a reliable reservoir of biogenic strontium, dentine is affected by diagenesis and contamination from the burial environment [56]. In fact, the original strontium isotope composition of dentine may vary post mortem up to 100% during the biochemical exchange with the soil. Therefore, although we cannot measure the exact proportion of variation, dentine’s values may be useful in a comparison to available strontium baselines for the area of the site.”

L136. 26 cremation – 3 infants = remaining 23 burials. Not 24.

Thanks, that was a misunderstanding, as one burial is multiple. 24 are the individuals, not the burials. Therefore, to clarify we rephrased as follows: “We sampled fragments of non-erupted tooth enamel in three cases (infant cremations nn. 350, 380, 449) and for the remaining 24 cremated individuals we used a fragment of the otic capsule of the pars petrosa (petrous portion) of the temporal bone, selecting those individuals with both preserved.”

L137-138. ‘selecting those individuals with both preserved’. What do the authors mean? Both PP preserved? If so, why? Why would you only select those individuals that have a left and a right PP preserved in their cremated remains?

We stated it in the “Materials” section. Strontium isotope analysis on cremated specimen is a destructive method and we did not want to compromise future investigations. It is possible that other isotopes will be more used and methods calibrated. 

L143-144. I disagree with the fact that the authors find it ‘reasonable to assume’ that the mothers did not migrate during the first 2 years of the child’s life. Archaeological and historical Sr data on dental enamel show childhood mobility in (pre)historic populations. It is a dangerous assumption to be made, but one that the whole concept of Sr isotope analysis in archaeology is based upon. However, it is never ‘reasonable to assume…’ if you don’t have archaeological or historical data to verify that assumption.

L143-144. This sentence is taken from Cavazzuti et al. 2019. I noticed quite some sentences that have been copy pasted or minimally modified from other papers…. (plagiarism: carefully check the manuscript!).

Thanks for the comment. I (Claudio Cavazzuti) might have put a few words about methodology taken from my previous papers published in the same journal (PLOS ONE). As I am the author and the copyright is of the same journal, I assume there is no big problem of plagiarism, also because it is not result or discussion section. Anyway, we changed the words and rephrased as follows: “It is seems scarcely probable that, in most cases, the infant’s mother/wet nurse did not change over the duration of breastfeeding, did not undertake long journeys, and did not consume exotic foods, which are not documented from the archaeological, archaeozoological and palaeobotanical record of Bronze Age Hungary [60].”

L149. Reference 68 does not refer to PP research. The authors should carefully check all references and use primary references instead.

L149. The Veselka et al. paper should be mentioned here as well.

Thanks for spotting this out, we removed ref. 68 and added Veselka et al. paper.

L153. The data is Ref 45 do not show the reliability of the PP, as the PP data are compared to that of the M2 that mineralises after the PP stops remodelling.

We agree and rephrased as follows: “Two tests for verifying the reliability and consistency of the pars petrosa results, were performed in one former contribution [46] and gave positive results, thus confirming the possibility of estimating the individual’s provenance also among cremations.”

L170. 2nd dentine is laid down in the pulp chamber. The authors sampled the outer surface of the root. And only a few mg. How can 2nd dentine be of interest for this study?

We must consider that molar roots are quite thin, compared to other teeth. Cremated dental roots are also a bit damaged by fire. Therefore, it is possible that part of the secondary dentine has been drilled.

L179. There are many more factors contributing to the turnover rate of bones (biological sex, health, age, location within the bone, stress, etc.). It is therefore almost impossible to predict the turnover rate of the fragment of bone selected for this study. Moreover, IF the average bone remodelling rate is estimated to be 3-8 years, the femur does not reflect the last few year of life, but the last 10 to 30 years (i.e., decades) of life.

We rephrased as follows: “The adult femur combines the isotopic signal of the formation phase and an average of the last years of life. In fact, once growth is complete, the bone is continuously remodelled through a cycle of resorption and apposition. The rate of bone remodelling is estimated to be from 3 to 8% per year, and this rate decreases with advanced age, with differences between skeletal parts [71–73].”

Materials in general: the text should be restructured, and the authors should systematically refer to all materials sampled. The text as in its current form is a hybrid text with sampled material and background information.

We tried to restructure the section as the Reviewer requested. If it is not a problem, we would prefer to maintain the information about dentine and bone physiology in this section to facilitate the reader in the understanding of our complex sampling strategy.

METHODS: SR ISOTOPE ANALYSIS

L217-226. No references are provided. 

We added Bentley, 2006 as one of the most cited review on the topic.

L231. “human bone and dentine can be used as baseline”. This approach is based on the very dangerous assumption that the bone and dentine samples are 100% diagenetically alterated. This text requires additional information.

We rephrased as follows: “If environmental samples are not available, or contaminated by modern sources [e.g. 82], archaeological fauna and human bone/dentine can be used to support local baselines, although, especially the latter, is not an ideal option and must be considered with caution, as diagenesis may alter to variable degree original strontium signatures.”

L238-241. This text is copy pasted from Cavazzuti et al., 2019 (plagiarism). Rewrite.

We rephrased as follows: “Concerning cremated individuals, petrous portions were sampled applying Jorkov et al.’s method [83]: the petrous bone was drilled at a 90˚ angle into the otic capsule (0.5–0.8 cm of depth), between the internal acoustic meatus and the subarcuate fossa with a low speed (2-mm diameter) drill, and a small fragment of bone was isolated.”

L234. This technique (sr isotopes) has been applied in archaeology for 30+ years. The technique itself is much older.

We rephrased as follows: “This technique has been in use for several decades and is now a common tool in mobility studies.”

L247. 30 ul columns? This seems incorrect to me.

It is correct. We currently employ in our lab 30 ul teflon columns with a reservoir of ca. 1 ml, consuming ca. 13 mg of dry Sr resin per sample. Almost all the works from the Modena laboratory of the last 3 years employed these columns (see e.g. Lugli et al., 2018, 2019; Weber et al., 2018; Argentino et al., 2019). The method was routinely employed by the Lamont-Doherty Earth Observatory at the Columbia and brought to our lab by Anna Cipriani.

L259. Please provide you blank data.

Added to the text as follows “The whole procedure was conducted in the clean lab of the Department of Chemical and Geological Sciences (University of Modena and Reggio Emilia), with a Sr blank typically lower than 100 pg. In this specific case, Sr laboratory blank was below 60 pg.”.

METHODS: GEOLOGY OF CENTRAL HUNGARY AREA AND BIOLOGICALLY AVAILABLE STRONTIUM ‘ISOSCAPE’

L307. If I understand correctly, results are given in this section? Results should be presented in the results section. I would include a paragrapgh about the additional baseline dat ain the results section, strengthening the range you presented in this section.

We moved the section at the beginning of the “results” paragraph.

RESULTS OF THE SR ANALYSIS

L414-415. Please provide the data for both females.

Done.

L427. You literally have one statistical outlier in phase 1. This is no convincing evidence of a “progressive reduction of mobility throughout the phases”, but overinterpretation of the data.

We rephrased as follows: “Moreover, the individuals dated to the second phase appear all compatible with local baselines (S1 Fig), and no significant differences can be observed comparing burials with different categories of grave goods (S2 Fig).”

L433. 1st Dentin is unable to remodel, it never remodels. Only 2nd and 3rd dentine are formed/added during life (and secondary dentin does remodel), but the primary dentine does not remodel. The samples taken for this study are not selected from the pulp cavity, hence the data ate probably not influenced at all by secondary dentine.

See comment L170

L436. No, not the foetus PP which you also refer to.

The foetus PP yields a value of 0.70932 which appears consistent with the SSM’s strontium signatures (0.7090-0.7097). 

L439. Be cautious. The isotopic differences between the samples are very very small and intra-individual isotopic differences do occur. You should avoid overinterpreting the data.

Also considering Reviewer’s 1 comments we changed as follows: “The data, therefore, suggest that the individual 241a (adult female) may have moved to Szigetszentmiklós during late childhood or early adolescence, sometime between the ages of 8 and 13. The slight increase of isotopic ratio that occurred between the remodelling of the femur cortical and the formation of 241b’s petrous bone might indicate movements in the final months or weeks of her life or, more plausibly, a slight change in the diet, in terms of the nature or provenance of the ingested food, which obviously also changed seasonally among farming communities of the Bronze Age.”

L 44. Same here. The difference between the pins extremely limited, also taken the ±2SE into account. These data do not necessarily point toward 2 different regions of origin.

Also considering Reviewer’s 1 comments we changed as follows: “Concerning the two bone pins/needles, the bone material of pin/needle 2 appears local, while bone/pin needle 1 shows a less radiogenic isotopic composition, slightly outside the local range. Different provenances may be supposed, but not without doubts. Despite the suggestive progression of the 87Sr/86Sr values of different tissues towards the local range, as we stated above, the difference between 241a’s petrous bone and local baselines is not so large (~0.0002) to assess an allochthonous provenance with certainty. For these reasons, in the next paragraph we will discuss the archaeological evidence (associated grave goods), in order to add further elements to our interpretative framework.”

Results in general: the authors tend to overinterpret the data. Moreover, the one datapoint of interest, the 0.7129 is not identified (male/female. Etc etc). Moreover, a statistical analysis of the data is absent that would probably underline the homogeneity of the data (except one). The interpretation of the possible non-local origin of 241 (0.7088) could also be strengthened using statistics.

We acknowledge that in the first version of this manuscript we did not provide a strong statistical evaluation of the data and we thank the Reviewer for the suggestion. We thus performed two outlier tests using the 3MAD approach and Tukey’s interquartile ranges. We detected two possible outliers (the two most radiogenic individuals of the dataset). ID 241 was not detected as outlier by neither method. However, by looking at the baseline interval, ID 241 is not compatible with the local samples. Anyhow, we decided to tone down our interferences on the likely non-local origin of 241. This, in turn, highlights the fact that we still lack solid methods for local baseline determination, especially for those sites with narrow isotopic/geologic variability, typically in the alluvial plains.

DISCUSSION

L467. No, the data are consistent with local, but the individuals could still be of a non-local origin.

We rephrased as follows: “In general, most of the individuals sampled from Szigetszentmiklós-Ürgehegy appear compatible with local strontium signatures. We cannot exclude the possibility that some of these might come from a distant or even remote region, but in that is the case, this is not detectable at either an isotopic or an archaeological level. Some patterns are nonetheless revealed by the integrated analysis of archaeological, osteological, and biogeochemical data.”

L468-470. Indeed, so don’t state that they “appear largely local”

See previous comment.

L478. Reference?

Added.

L480. In this study only 3 (!!) inhumed individuals are included. One cannot and should never ‘weaken a hypothesis’ based on only 3 datapoints. Try not to overinterpret the data,

Here, we also refer to Érd-Hosszúföldek and Százhalombatta-Belső-újföldek, so the total number of inhumations in the area is larger. However, we accepted the suggestion and rephrased as follows: “Hence, despite the fact that the introduction of inhumation in the Vatya area around 1500 BC has been traditionally associated with the introgression of newcomers from north-west, the current isotopic data does not provide clear evidence of the presence of immigrants, at least for these specific contexts. We cannot, nonetheless, exclude the possibility that some of these individuals might have come from distant regions characterised by similar geolithologies and, therefore, strontium isotope compositions.”

L486. Now the outlier is identified in the text, 243. I agree that 243 is of non-local descent. The data for 241, however, nis not convincing.

We rephrased accordingly: “The 87Sr/86Sr value obtained from her petrous bone (0.70888) falls outside the local range (0.7090-0.7097). The difference is not so large to estimate an allochthonous provenance with absolute certainty, but the overall distribution of data shows that her strontium signature diverges from the major concentration of individuals (0.70906-0.70956) (see density plot, Fig. 8). Therefore, it seems plausible to hypothesize that she was not originally a member of the indigenous community and moved to Szigetszentmiklós, most likely between the ages of 8 and 13, also in light of the archaeological materials.

Among the ornaments that were part of her funeral dress, the two bone needles were certainly on the pyre, since they show the typical, white-calcined appearance of cremated bone. The gold ring and the bronze neck-ring do not show any clear trace of being exposed to the fire, although gold and copper/bronze melt at around 1000°C the pyre might not have been fully reached such temperatures. More likely, they were added as grave goods during the deposition of the ashes in the urn. 

Considering that the woman might have come from outside, it seems relevant to analyse the distribution of these artefacts in a broader geographical context and to explore whether there might be any compatibility between the potential provenance of the individual and of her grave goods.”

L498. The authors should first check the statistically significant. Without this data, the interpretation in L498 seems incorrect.

After statistical analyses, we toned down and refined our interpretation.

We also performed a statistical analysis on the data in order to apply some caution in the following interpretation. Therefore, we added a paragraph in the Methods and Results sections.

Methods: “To statistically detect possible non-local individuals, we employed two approaches: the median absolute deviation from the median (MAD) and the Tukey’s fences method. Concerning the MAD, we calculated the 3MADnorm as reported in Lightfoot and O’Connell (2016) and Leys et al. (2013), namely multiplying by 3 the MAD and scaling it to a b value. This latter is usually assumed as a constant (1.4826) for normal distributions, neglecting outlier-induced abnormalities (Leys et al., 2013). Excluding outliers though a Grubb test (p < 0.05; 0.71209 is an outlier), our data appears as normally distributed (Shapiro-Wilk test; W = 0.954; p = 0.105), hence a b value = 1.4826 has been used as scaling factor for the MAD. Tukey’s interval has been calculated as Q1-1.5(IQR) and Q3-1.5(IQR), where IQR is the interquartile range (Tukey, 1977). A two-tailed T-test was also performed searching for group differences between cremated vs. inhumated individuals. We acknowledged that the small number of inhumated individuals (n = 3) may impact the prediction power of the test. All the tests have been performed manually using either Microsoft Excel or MATLAB.”

Results: ““Statistical results are summarized in Table 7. Both employing 3MADnorm or Tukey’s IQR [88–90], only two individuals are detected as outliers, namely n. 215 (87Sr/86Sr =0.70956) and n. 243, the most radiogenic of the dataset (87Sr/86Sr=0.71209). N. 215, however, fall within the local strontium signatures (Szigetszentmiklós=0.7091-0.7095; immediate hinterland=0.7090-0.7097). Individual n. 241a (the lowest radiogenic; 87Sr/86Sr=0.70888) is outside the local range but lays at the lower edges (but still within) of the statistical intervals (3MADnorm or Tukey’s IQR).

Table 1. Summary statistics for 87Sr/86Sr ratios of bone and tooth specimens.

Sample size 41

MIN 0.70888

MAX 0.71209

Mean 0.70926

2SD 0.00094

Median 0.70917

3MADnorm 0.00033

IQR 0.00015

Tukey’s fences 0.70887-0.70948

No significant differences have been observed between cremated and inhumated individuals, in terms of Sr isotope ratio (two tailed T-test, p = 0.75). However, removing the two detected outliers (i.e. likely non-locals, see above), the three inhumated appears significantly different from the local cremated individuals (T-test, p = 0.016).“

L509-563

Now the artefacts are discussed in great detail. This part of the research is, however, not properly introduced in the introduction. Should it indeed be included in the text, or transferred to the supplementary data? I do see the link between the artefacts and 241, but I would suggest to restructure the text completely, and focus on 241 and her grave goods in a dedicated paragraph.

We accept this comment, and we acknowledge the point made by the Reviewer 2, but we would prefer to maintain the archaeological observations in the paper instead of the supplementary information, in order to have a real interdisciplinary, comprehensive reflection, which may integrate multiple evidence for the general interpretation of that unique burial. We also added a sentence in the Introduction, to specify this task.

“Our main goals are a) contributing to the understanding of cremation practices of the Early and Middle Bronze Age in Hungary; b) exploring mobility patterns among inhumations and cremations, as well among individuals of both sexes and different age class, through strontium isotope analysis; c) applying a ‘biographic’ approach, namely targeting different human cremated tissues for strontium isotope analysis, to one particularly relevant case, that of burial n. 241, which contains multiple individuals and a variety of prestige grave goods. These latter have also been analysed from the point of view of geographical and contextual distribution, in order to integrate osteological and isotopic data in the broader archaeological framework.”

L575. Be careful not to overinterpret. Maybe the ornaments have been in the family for decades and do not reflect the female’s origin, but that of her tribe/culture/family. This complexity of prehistoric populations and their social interactions and family links cannot be easily solved and alternative interpretations should be reflected upon as well.

We agree and therefore added: “However, we cannot exclude that the Ösenring could belong to another member of the family, of the social group, or to an ancestor.”

CONCLUSIONS

L590. Overinterpretation of the data. Please provide statistical evidence.

We performed a T-test on the two set of data, although we acknowledge that the sample size for the inhumated individuals is quite small and certainly influenced the power of the test. The results are significant only excluding the non-local individuals among the cremated. We amended our inferences based on this evidence. 

L594. aDNA on well cremated remains?

We were referring to inhumations. Perhaps in the current form, the text is clearer.

L595 - end. Again, suggestive. The data are not strong enough to provide solid evidence for this conclusion.

We rephrased as follows: “Our study demonstrates that some of the non-cremated individuals at Szigetszentmiklós-Ürgehegy appear slightly different from an isotopic perspective, compared to those who were cremated. Inhumations are nonetheless compatible with the local baselines. Further strontium/oxygen/lead isotopes and aDNA analysis might further elucidate the issue of the possible introgression of newcomers from central Europe in the Late Vatya period.

The isotopic evidence also suggests that the community of Szigetszentmiklós-Ürgehegy was largely patrilocal and practiced exogamy to a certain extent.

The deceased from burial n. 241a seems to fit within this general picture and emphasises the importance of the movement of high-status female individuals. Strontium isotope analysis of her bone/tooth tissues shows that she might have moved from outside the area to Szigetszentmiklós when she was between 8 and 13 years of age, in the period of the menarche, and therefore at the beginning of the potentially fertile stage of her life. The geographic distribution of the objects which were part of her mortuary parure, and the Ösenring in particular, may indicate that she had origins in Southern Moravia, Lower Austria or in the upper Danube valley, for example in the Pre-Alpine Bavarian lowlands. All these areas show 87Sr/86Sr baseline which are consistent with the values obtained on her petrous bone. Although archaeological and biogeochemical data converge rather significantly, we must remark that other areas of provenance could not be excluded.”

Minor (textual) changes

Abstract

L29. The comma (,) should be a semi-colon (;)

L32. Use the oxford comma after ‘apogee’

L37. Here “our results” are presented, but ideally in an abstract, the applied methodology should be mentioned as well. This is absent in the abstract and should be added.

L41-43. The abstract should mention the results, a synopsis of the work executed. This section mentioned the work (Sr analysis) that have been done, but fails to mention the (interpret) data.

L328. Comma before ‘with’.

L332. Old= of age

L474. Isotopic= isotope

Thanks, we changed the text accordingly. The abstract, in particular, changed as follows: “In this study, we present strontium isotope data for the analysis of human mobility carried out on 29 individuals (26 cremations and 3 inhumations) from Szigetszentmiklós-Ürgehegy, one of the largest Middle Bronze Age cemeteries in Hungary. The site is located in the northern part of the Csepel Island (a few kilometres south of Budapest) and was in use between c. 2150 and 1500 BC, a period that saw the rise, the apogee, and, ultimately, the collapse of the Vatya culture in the plains of Central Hungary. The main aim of our study was to identify variation in mobility patterns among individuals of different sex/age/social status and among individuals treated with different burial rites using strontium isotope analysis. Changes in funerary rituals in Hungary have traditionally been associated with the crises of the tell cultures and the introgression of newcomers from the area of the Tumulus Culture in Central Europe around 1500 BC. Our results show only slight discrepancies between inhumations and cremations, as well as evident differences between adult males and females.

The case of the richly furnished grave, n. 241, is of particular interest. The urn contains the cremated bones of a non-indigenous adult woman and two 7 to 8-month-old foetuses, as well as prestigious goods. Using 87Sr/86Sr analysis of different dental and skeletal remains, which form in different life stages, we were able to reconstruct the potential movements of this high-status woman over almost her entire lifetime, from birth to her final days. 

Our study confirms the informative potential of strontium isotopes analyses performed on cremated remains. From a more general, historical perspective, our results reinforce the idea that exogamic practices were common in Bronze Age Central Europe and that kinship ties among high-rank individuals were probably functional in establishing or strengthening interconnections, alliances, and economic partnerships.”

---

## [Decision Letter · Decision Letter 1]

3 Jun 2021

PONE-D-20-34426R1

Human mobility in a Bronze Age Vatya ‘urnfield’ and the life history of a high-status woman

PLOS ONE

Dear Dr. Cavazzuti,

Thank you for submitting your manuscript to PLOS ONE. After careful consideration, we feel that it has merit but does not fully meet PLOS ONE’s publication criteria as it currently stands. Therefore, we invite you to submit a revised version of the manuscript that addresses the points raised during the review process.

The thorough revision conducted by the authors was highly appreciated. Reviewer #2 points to some minor revisions, to which I would add the following remarks. Names of site and rivers are hardly readable on the map of figure 1 and a contextualization map should be added as embedded picture. Please add some scale reference for figure 2, 4 and 5. Does Figure 5 show the bones of both foetuses (see caption)? Could it be combined with Figure 4? In figure 7, in addition to add axis lines, please have the number given in a consistent way (0.7090 instead of 0.709 along the y axis of the bottom graph). Figures 8 and 9 present a y axis with numbers including only three decimals, while 4 (even 5 in Figure 10) decimals are given elsewhere. I suggest transferring Figure 11 and 12 in the supplementary data (please note that the name rivers are again difficult to read) and to add an illustration (picture, drawing) of bronze neck-ring and gold ring as embedded picture (e.g. top right of the map) to have a better visualization of the conveyed information. After the authors send a new version integrating these last minor revisions, I will be glad to accept the contribution for publication.

We look forward to receiving your revised manuscript.

Kind regards,

Dorothée Drucker

Academic Editor

PLOS ONE

Journal Requirements:

Reviewers' comments:

Reviewer's Responses to Questions

**Comments to the Author**

1. If the authors have adequately addressed your comments raised in a previous round of review and you feel that this manuscript is now acceptable for publication, you may indicate that here to bypass the “Comments to the Author” section, enter your conflict of interest statement in the “Confidential to Editor” section, and submit your "Accept" recommendation.

Reviewer #1: All comments have been addressed

Reviewer #2: All comments have been addressed

2. Is the manuscript technically sound, and do the data support the conclusions?

Reviewer #1: Yes

Reviewer #2: Yes

3. Has the statistical analysis been performed appropriately and rigorously? 

Reviewer #1: Yes

Reviewer #2: Yes

4. Have the authors made all data underlying the findings in their manuscript fully available?

Reviewer #1: Yes

Reviewer #2: Yes

5. Is the manuscript presented in an intelligible fashion and written in standard English?

Reviewer #1: Yes

Reviewer #2: Yes

6. Review Comments to the Author

Reviewer #1: I thank the autors for taking into account the comments of both reviewers. I have no further comments and look forward to seeing this paper published.

Reviewer #2: I am glad to see that the authors took the feedback and suggestions well and incorporated them into the revised manuscript. The text has certainly improved as a result of the adjustments made by the authors: more readable, but also the quality of the data and its interpretation improved significantly. Except for a few minor comments (see below), I recommend the paper to be accepted in its current form

Few minor comments:

- the abstract mentions 29 individuals, while the MM section refers to 30.

- Supplementary data 14C: could you add the sample ID of the 14C laboratory?

- Supplementary data 14C: chronotipology � chronotypology

- Supplementary data 14C: age (in years). Ad = ?

-

Few suggestions

- Fig 7: density plot: could you add the X-axis?

- Fig 7: graph: could you put 87 and 86 in superscript and replace the commas (,) with periods (.)

- Fig 8-9-10: ditto in superscript

- Suppl.3-4: ditto in superscript

7. PLOS authors have the option to publish the peer review history of their article (what does this mean?). If published, this will include your full peer review and any attached files.

Reviewer #1: No

Reviewer #2: No

---

## [Author Response · Author response to Decision Letter 1]

22 Jun 2021

Dear Editor and Reviewers,

we are very grateful to you for the attention given to our manuscript. We changed the figures, tabs, caption and text according to your suggestions.

Also on behalf of my co-authors, thanks again for your efforts,

Yours,

Claudio Cavazzuti

EDITOR

Names of site and rivers are hardly readable on the map of figure 1 and a contextualization map should be added as embedded picture. 

- We enlarged river labels.

Please add some scale reference for figure 2, 4 and 5.

- Done.

Does Figure 5 show the bones of both foetuses (see caption)?

- We changed the caption to “Bones attributable to both foetuses (n. 241b and 241c).”

Could it be combined with Figure 4?

- Done. We changed all the following figures and reference to figures in the text and captions.

In figure 7, in addition to add axis lines, please have the number given in a consistent way (0.7090 instead of 0.709 along the y axis of the bottom graph).

- Done.

Figures 8 and 9 present a y axis with numbers including only three decimals, while 4 (even 5 in Figure 10) decimals are given elsewhere.

- Done. We made them homogenous.

I suggest transferring Figure 11 and 12 in the supplementary data (please note that the name rivers are again difficult to read) and to add an illustration (picture, drawing) of bronze neck-ring and gold ring as embedded picture (e.g. top right of the map) to have a better visualization of the conveyed information.

- Done. Unfortunately, we do not have the possibility of changing the font size of river names, but as it is a digital version, one can zoom in, in case.

REV2

The abstract mentions 29 individuals, while the MM section refers to 30.

- Thanks. We changed the MM section as follows: “We collected a total of 41 samples from different human tissues included in 29 burials (3 inhumations and 26 urn cremations; burial n. 241 includes three individuals, two of them analysed), as well as from 3 bone pins/needles from burials n. 215 and 241a (Table 1).”

- 

Supplementary data 14C: could you add the sample ID of the 14C laboratory?

- Yes, we did it.

Supplementary data 14C: chronotipology � chronotypology

- Changed

Supplementary data 14C: age (in years). Ad = ?

- We changed to “20-40”

Few suggestions

Fig 7: density plot: could you add the X-axis?

- Done

 Fig 7: graph: could you put 87 and 86 in superscript and replace the commas (,) with periods (.)

- Done

Fig 8-9-10: ditto in superscript

- Done

Suppl.3-4: ditto in superscript

- Done

---

## [Editor Report · Decision Letter 2]

28 Jun 2021

Human mobility in a Bronze Age Vatya ‘urnfield’ and the life history of a high-status woman

PONE-D-20-34426R2

Dear Dr. Cavazzuti,

We’re pleased to inform you that your manuscript has been judged scientifically suitable for publication and will be formally accepted for publication once it meets all outstanding technical requirements.

Kind regards,

Dorothée Drucker

Academic Editor

PLOS ONE
---

## [Editor Report · Acceptance letter]

7 Jul 2021

PONE-D-20-34426R2 

Human mobility in a Bronze Age Vatya ‘urnfield’ and the life history of a high-status woman 

Dear Dr. Cavazzuti:

I'm pleased to inform you that your manuscript has been deemed suitable for publication in PLOS ONE. Congratulations! Your manuscript is now with our production department. 

Kind regards, 

on behalf of

Dr. Dorothée Drucker 

Academic Editor

PLOS ONE